# The Mitochondrial Connection: The Nek Kinases’ New Functional Axis in Mitochondrial Homeostasis

**DOI:** 10.3390/cells13060473

**Published:** 2024-03-07

**Authors:** Fernanda L. Basei, Ivan Rosa e Silva, Pedro R. Firmino Dias, Camila C. Ferezin, Andressa Peres de Oliveira, Luidy K. Issayama, Livia A. R. Moura, Fernando Riback da Silva, Jörg Kobarg

**Affiliations:** Faculty of Pharmaceutical Sciences, University of Campinas, Campinas 13083-871, Brazil; fbasei@unicamp.br (F.L.B.); pedrofirminodias@gmail.com (P.R.F.D.);

**Keywords:** Nek kinase family functions, mitochondrial homeostasis, cell cycle, cellular signaling

## Abstract

Mitochondria provide energy for all cellular processes, including reactions associated with cell cycle progression, DNA damage repair, and cilia formation. Moreover, mitochondria participate in cell fate decisions between death and survival. Nek family members have already been implicated in DNA damage response, cilia formation, cell death, and cell cycle control. Here, we discuss the role of several Nek family members, namely Nek1, Nek4, Nek5, Nek6, and Nek10, which are not exclusively dedicated to cell cycle-related functions, in controlling mitochondrial functions. Specifically, we review the function of these Neks in mitochondrial respiration and dynamics, mtDNA maintenance, stress response, and cell death. Finally, we discuss the interplay of other cell cycle kinases in mitochondrial function and vice versa. Nek1, Nek5, and Nek6 are connected to the stress response, including ROS control, mtDNA repair, autophagy, and apoptosis. Nek4, in turn, seems to be related to mitochondrial dynamics, while Nek10 is involved with mitochondrial metabolism. Here, we propose that the participation of Neks in mitochondrial roles is a new functional axis for the Nek family.

## 1. Introduction

Neks (NIMA-related kinases) are serine/threonine kinases that were initially categorized as cell cycle-related kinases, like its fungi-orthologous NIMA (Never-In-Mitosis A (NimA)), which is essential to mitosis entry in that organism [1,2,3]. However, growing evidence points to additional roles of the eleven human kinases in addition to the cell cycle control.

The kinase domain is conserved among the Nek family members, while the C-terminus, called the regulatory domain, is vastly variable, ranging from hundreds (Nek2 and Nek3) to thousands of residues (Nek1) containing coiled coils (Nek1, Nek2, Nek5, Nek9, Nek10, and Nek11), PEST sequences (Nek1, Nek2, Nek3, Nek9, Nek10, and Nek11), DEAD-box domain (Nek5), and Armadillo repeats (Nek10) [4,5,6,7,8]. This structural diversity clearly suggests additional roles for these proteins beyond that of the fungal orthologue.

Indeed, participation in DNA damage response, alternative splicing, and cilia formation has already been described for several members of the Nek family (for review, see [9,10,11]) [12,13]. A mitochondrion-related function was first shown for Nek1 [14,15], and from that, it was extended to other members: Nek5 [16,17,18], Nek10 [19], Nek4 [13,20], and Nek6 [20,21], so far.

Mitochondria are fundamental to cellular energy supply, lipid metabolism, calcium homeostasis, and cell death. Mitochondria are well-structured organelles, with a double membrane and a matrix containing its own genetic content, the mtDNA. As mitochondria cannot be formed de novo, their correct distribution during cell division is also key for cell functionality. In this context, Neks could be an important link between the nuclear events and the distribution of these organelles during cell division.

Mitochondrial functions are regulated by mitochondrial morphology, protein composition, and mtDNA content and integrity. Our interactome studies on Nek1 [21], Nek4 [13], Nek5 [16], Nek10 [18], and Nek6 [22] revealed several mitochondrial or mitochondrion-related proteins. Based on that, we suggest five Nek family members that participate in the regulation of distinct mitochondrion-related functions, such as mitochondrion composition/structure, metabolism, mtDNA maintenance, cell stress response, and autophagy (Figure 1).

In this review, we focus on the new roles of Neks related to distinct aspects of the mitochondrial function.

## 2. Nek Interactomes Reveal Mitochondrion-Related Functions

We found the first pieces of evidence of Neks acting in mitochondrial roles from our interactome studies. The first Neks to be assigned to several mitochondrial interactors were Nek4 and Nek5. Concomitantly, Chen and colleagues [14] demonstrated that Nek1 phosphorylates VDAC1 (voltage-dependent anion-selective channel 1) on residue S139. Subsequently, the interaction among Neks and some mitochondrial proteins was confirmed (COX11, MTX2, and LONP1 for Nek5 and GLUD1 for Nek10) [16,17,19]. Furthermore, an indirect interaction with proteins not located in mitochondria but fundamental for mitochondrial functions, like Dynamin-like-1 protein (DNM1L gene, usually named DRP1), DRP1, was described for Nek4 [20]. Considering our data from interactome studies, we can observe that Neks are significantly implicated in different mitochondrion-related processes. The Nek1 interactome showed a significant number of proteins related to apoptosis and autophagy (Figure 1 and Table 1), while the Nek4 interactome showed enrichment in proteins linked to mitochondrial structure (Figure 1) and stress response (Table 1); Nek10 was related to mitochondrial respiration (Table 1). Curiously, the Nek5 interactome pointed to the enrichment of autophagy-related genes, a process that was found to be enriched in two different studies, using different cell lines (Table 1). Although the enrichment analysis for the Nek6 interactome did not show mitochondrion-related processes, some stress response proteins, such as thioredoxin-dependent peroxide reductase, PRDX3, the transcription factor RelB, (RELB), the matrix metalloproteinase-2 (MMP2), and the von Willebrand factor A domain-containing protein 8 (VWA8), and the mitochondrial ribosomal proteins MRLP19 and MRLP28 were identified (Figure 1).

Most of the proteins present in the mitochondrion are encoded in nuclear DNA. These proteins usually contain a mitochondrial localization signal (MLS) or targeting signal (MTS). The canonical signal is a short peptide that contains positively charged basic residues (15–70 aa), usually at the N-terminus of the protein [25,26,27]. However, MTS can be located in the middle region (ADP/ATP carrier) or at the C-terminus (VDAC) of the amino acid sequence [28,29,30]. Proteins with a C-terminal MTS usually reach mitochondrial transporters, complexed with chaperones Hsp70/90. While the mitochondrial import receptor subunit (TOMM20) recognizes the N-terminal mitochondrial targeting sequence, TOMM70 can recognize internal or C-terminal targeting sequences [27,31]. Usually, proteins directed to the mitochondrial matrix contain presequences that are cleaved by a peptidase in the inner mitochondrial membrane (IMM).

Using available predictors (MitoPro II, Target 2.0, and MitoFates), we found that most Neks indeed do not contain a presequence or a mitochondrial transient peptide, suggesting that they are not imported to the mitochondrial matrix (Table 2). The three predictors use machine learning and train the model mainly with classical proteins (from different species) possessing mitochondrial presequences. Features from signal peptide (SP), such as positively charged residues, hydrophobicity, and amino acid residues present around the cleavage site were considered for prediction [32]. For comparison, we parallelly analyzed Mitofusin 1 (Mfn1), a mitochondrial outer membrane (MOM) protein, autosomal-dominant optic atrophy (OPA1), a mitochondrial inner membrane (MIM) protein, and the transcription factor A (TFAM), a mitochondrial matrix protein, as control proteins for these analyses.

Indeed, only Nek11 showed a high probability for mitochondrial import (0.749), at comparable levels of TFAM (0.919) and OPA1 (0.998), according to MitoProtII. However, an important point to be considered is that the available predictors only consider the N-terminus of the protein and as mentioned before, growing evidence indicates that the signal for mitochondria localization can also be present internally or at the C-terminus of the protein. Curiously, Nek5, the isoform 1 of Nek4 and Nek2 (especially isoform C) showed higher probability scores among Nek family members for mitochondrial localization based on the MitoProt II algorithm [33]. These scores (0.149, 0.144, and 0.248, respectively) are comparable to the probability score calculated for Mfn1 mitochondrial import (0.204) (Table 2). Nek4 is located at mitochondrial crude fractions and more specifically, at the microsome, but no localization at pure mitochondrial fractions was detected in HEK293 cells [20]. Nek6, Nek7, Nek9, and Nek10 did not show a recognition motif for TOM20. It is interesting to note that Nek5 was found to interact with Metaxin 2 (MTX2) [16], a MOM protein that participates in mitochondrial protein import and is a component of the sorting and assembly machinery of the outer membrane (SAM/TOB) required for β-barrel protein insertion at MOM [34,35,36,37].

Nek10 is the most divergent Nek regarding domain organization. The main isoform of NEK10 is a serine/threonine/tyrosine kinase of 133 kDa. The kinase domain is present in the center of the protein. The coiled coil region is located near the kinase domain, and there are four armadillo motifs at the amino-terminal regulatory domain, which probably act in protein–protein interactions [9,19,38]. Considering the latter and the assays conducted by our group, one can conclude that a specific isoform of Nek10 is located in mitochondria [19] or that the MTS is located in a region other than the N-terminus. Further analysis using predictors that consider the entire sequence or a specific Nek10 isoform after its characterization could clarify this absence of mitochondria localization prediction.

So far, there is no information regarding the mitochondrial localization of Nek6, and its effect on mitochondrial functions could be related to transcriptional regulation or interactions with proteins important to that organelle. In the next sessions, we provide in-depth information regarding Neks 1, 4, 5, 6, and 10 in different mitochondrial functions and comment on the potential relation with their interaction partners.

**Table 1 cells-13-00473-t001:** Enrichment of the biological process with mitochondrion-related genes for Neks.

	Biological Process (BP, Mitochondrial Related Genes)							
	Autophagy		Apoptosis		Stress Response		Respiration-Related	
	% of Genes	*p*-Value	% of Genes	*p*-Value	% of Genes	*p*-Value	% of Genes	*p*-Value
Nek1	4.3	1.3 × 10^−5^	8.1	1.5 × 10^−4^	6.2	9.6 × 10^−3^	-	-
Nek4	-	-	-	-	1.9	3.0 × 10^−3^	0.6	6.5 × 10^−2^
Nek5	3.5	2.2 × 10^−3^	-	-	3.5	1.2 × 10^−4^	-	-
Nek10	-	-	-	-	-	-	13	7.3 × 10^−3^

The whole list of protein interactors for Nek1, Nek4, or Nek5 was added to DAVID [39], and the percentage of genes present in the list that correspond to the specific BP and *p*-value for the biological process related to mitochondrial functions are presented. For Nek10, the list with genes (encoding the interacting proteins) related to all mitochondrion-related functions was added to DAVID, and the predominant biological process is presented. In the case of Nek10, only the genes related to mitochondrial functions were used. The electronic transport (13% *p*-value: 7.3 × 10^−3^), TCA cycle (8.7% *p*-value: 2.9 × 10^−2^), and respiratory chain (8.7% *p*-value: 7.5 × 10^−2^) were the only three processes identified. The Nek6 gene list did not show any of these processes enriched.

**Table 2 cells-13-00473-t002:** Nek mitochondrial localization prediction.

	MitoProt II	Target 2.0	MitoFates	
	Probability	mTP	Probability of Presequence	TOMM20 Recognition Motif
Nek1	0.0356	0.000163	0.003	Yes
Nek2A	0.1593	0.000069	0.002	Yes
Nek2B	0.1367	0.000069	0.002	Yes
Nek2C	0.2477	0.000069	0.002	Yes
Nek3	0.0518	0.001566	0.005	Yes
Nek4.1	0.1438	0.019433	0.013	Yes
Nek4.2	0.0548	0.019433	0.013	Yes
Nek5	0.1493	0.000059	0.001	Yes
Nek6	0.0055	0.000293	0.01	No
Nek7	0.0044	0.000016	0	No
Nek8	0.1201	0.000873	0.075	Yes
Nek9	0.0242	0.000003	0	No
Nek10	0.0040	0.000013	0.001	No
Nek11	0.7487	0.153774	0.888	Yes
Mfn1	0.2037	0.000086	0.02	Yes
OPA1	0.9981	0.915521	0.509	Yes
TFAM	0.9190	0.964329	0.996	Yes

Amino acid sequences of canonical isoforms of Neks (or from characterized isoforms in the case of Nek2 and Nek4) were submitted to three predictors of mitochondrial localization. MitoProtII [39] accounts for the probability of import to mitochondria, considering the presence of a mitochondrial target sequence at the N-terminus of the protein and the cleavage site. Target 2.0 [32] evaluates the presence of the N-terminal mitochondrial transient peptide (mTP) that ranges from 0 to 1, i.e., not present to present with high confidence. The mTP is frequently found in mitochondrial matrix proteins. MitoFates [40] calculates the probability (also ranging from 0 to 1) of the presence of cleavable N-terminal presequences necessary for mitochondrial import and the TOMM20 recognition motif at the N-terminus. TFAM, OPA1, and Mfn1 were used as a control for matrix, IMM, and OMM proteins respectively.

## 3. Neks and the Mitochondrial Metabolism

The main role of mitochondria is to provide energy in the form of adenosine triphosphate (ATP) and guanosine triphosphate (GTP). The production of both ATP and GTP molecules occurs using mainly glucose, but amino acids and fatty acids can also serve as substrates of origin. Mitochondrial ATP production occurs through sodium pyruvate that enters the mitochondrial matrix and feeds into the tricarboxylic acid (TCA) cycle. In the electron transport chain (ETC), a proton gradient is generated, which in turn serves to propel the ATP synthase to produce ATP (for review, see [41]).

To produce ATP properly, the complexes of proteins that participate in oxidative phosphorylation (OXPHOS) must be organized in a well-structured membrane. In addition, the composition of the proteins that form the complexes must be maintained. Five multi-protein complexes participate in the electron transport chain: complex I (NADH-ubiquinone oxireductase), complex II (Succinate dehydrogenase, complex III (cytochrome c oxidoreductase), complex IV (cytochrome c oxidase), and the ATP synthase, complex V (for review, see [42]). Complex V comprises two multi-subunit complexes (F0 and F1) [43].

The TCA cycle is the central connection between glucose, lipid, and amino acid metabolism. The metabolic intermediates of glutamine, glutamate, leucine, and isoleucine synthesis/degradation also feed the TCA. The acetyl-oA from beta-oxidation can also be used in the TCA cycle. In addition, beta-oxidation produces NADH that can be used for ETC (for review, see [41]).

As a site of a variety of oxi-reduction reactions, the mitochondrion is the main source of reactive oxygen species (ROS) generation in the cell. ROS are generated from the reduction of oxygen to superoxide, hydrogen peroxide, and hydroxyl radical. ROS are important signaling molecules but can also impose danger to proteins and mtDNA integrity. The mitochondrion has a set of important enzymes that metabolize ROS; the superoxide dismutase (SOD2), glutathione peroxidase (GPX4), and peroxiredoxin (PRDX3) are the main antioxidant defense of mitochondria (for review, see [41]). Below, we discuss the role of Neks in respiration and mitochondrial metabolism.

### 3.1. Neks 1, 4, 6, and 10 Increase Mitochondrial Respiration, as Opposed to Nek5

Recent studies indicate a correlation between Nek1 and the metabolic function of the cell, adding some new information on Nek1 mitochondrial activity. The most recent paper published regarding this subject used mice models to study the well-known Nek1^kat2J/kat2J^ loss-of-function mutation on mouse embryonic fibroblast (MEF) cells to understand the mechanisms underlying some neurodegenerative disorders [44]. This study showed decreased levels of glucose transporter 1 (GLUT1) and consequently, reduced glucose uptake and downstream glucose metabolites such as glucose-6-phosphate and fructose-1,6-bisphosphotate and metabolic products of ^13^C_6_-labeled glucose, including ^13^C_3_-labeled phosphoenolpyruvate and pyruvate. As expected, oxygen consumption levels (tested through seahorse analysis) and levels of acetyl-CoA and ATP were also reduced (Figure 2). Interestingly, the lethality of Nek1^Kat2J/Kat2J^ (Nek1 deficient) in preweaning mice (around 2 weeks of age) can be reduced by the expression of a catalytically inactive RIPKinase1 or a ketogenic diet [44].

Using prostate cancer cell lines, LNCaP, and the epithelial salivary gland, HSG, Singh and colleagues (2020) [45] showed an increase in basal and ATP-linked respiration in cells overexpressing Nek1. The maximal capacity and spare capacity were also increased in Nek1-overexpressing cells [45].

Recently, our group, using HAP1 Nek1 KO cells, showed a decrease in complex I activity, which could be related to a decrease in the transcription of several complex I subunits. Nek1 KO cells also showed an increase in total and mitochondrial ROS [46] (Figure 3).

Since oxygen plays a key role in respiratory activity and metabolic functions, understanding the mechanisms behind oxygen metabolism is fundamental to comprehending mitochondrial activity. Hypoxia is a condition with low levels of oxygen, to which mitochondria respond by lowering their activity and balancing ATP consumption and production, thus preventing a bioenergetic collapse [47]. Under normal levels of oxygen (normoxia), prolyl hydroxylase domain proteins (PHD) and hydroxylate hypoxia-induced factors (HIFs) allowed the recognition of von Hippel–Lindau (VHL) proteins of the E3 ubiquitin ligase complex to ubiquitinate HIFs and promoted their proteasomal degradation. Under hypoxia conditions, PHDs are no longer hydroxylated HIFs, allowing the ligation to hypoxia response elements (HRE) in hypoxia-regulated genes until the hypoxia is resolved [48].

In this context, Nek1 has been shown to be a hypoxia-regulated gene through the interaction of HIF2-α with HRE located in the Nek1 gene promoter. Moreover, VHL can promote Nek1 degradation via ubiquitin-proteasome, supporting the idea of a hypoxia-regulated gene [48]. However, Nek1 has been shown to phosphorylate and interact with VHL and promote its proteasomal degradation, which indicates a positive feedback mechanism [49]. Although these mechanisms have been revealed, the clear correlation between them and mitochondrial activity is yet to be determined.

Lastly, Nek1 knockout HAP1 cells affected the expression of some key elements of mitochondrial functions and induced cellular and mitochondrial ROS. The transcriptome analysis through RNA sequencing revealed that some key elements of complex I had a reduced expression in Nek1 knockout cells, which can explain the increased cellular ROS (measured with DHE) and mitochondrial ROS (measured with mitoSOX) [46] since it has already been shown that the inhibition of complex I with rotenone, for example, induces ROS formation and mitochondrial DNA damage [50].

Furthermore, the first evidence suggesting a role of Nek4 in mitochondria came from our interactome studies, showing several proteins related to mitochondrial functions [13] (Figure 1). Recently, the role of Nek4 in regulating mitochondrial ATP production was demonstrated [20]. Nek4 increased basal and ATP-coupled mitochondrial respiration with increased ATP concentration, while Nek4 depletion had the opposite effect (Figure 2 and Figure 3). It was also shown that Nek4 expression and depletion alter the expression of some proteins from the respiratory complexes without altering overall mitochondrial mass. For instance, Nek4 overexpression is associated with a major expression of complex I protein, NDUFB8, while its knockdown leads to a decrease in UQCRC2 from complex III expression and in MTCOI from complex IV expression [20].

The overexpression and depletion of Nek4 resulted in an increase and a decrease in mitochondrial membrane potential (MMP), respectively. High mitochondrial ROS levels were observed in both conditions (overexpression and depletion), which, in the case of Nek4 knockdown, is due to a reduction in mitochondrial function. In the case of Nek4 overexpression, the high ROS can be explained by the high respiratory capacity and is compensated by a high antioxidant activity. Indeed, the levels of PRDX3 are higher in Nek4-overexpressing cells, and this is according to the reduced protein carbonylation observed in these cells [20].

Nek10 depletion also impairs mitochondrial respiration [19]. The maximal respiration capacity is increased in the control cells compared to Nek10-depleted HeLa cells. In Nek10-depleted HeLa cells, the ATP-linked OCR (oxygen consumption rate) is decreased related to the control cells (Figure 2 and Figure 3). In addition, the control cells showed higher spare capacity compared to Nek10-depleted HeLa cells. Furthermore, there was an increase in the proton leak in Nek10-depleted HeLa cells compared to the control cells. The control cells consumed more oxygen outside the mitochondria than depleted cells [19]. These results suggest that Nek10 could control OXPHOS. The loss of CII leads to ROS generation in cells, which is associated with cancer and neurodegenerative diseases [51].

Additionally, Hanchuck and colleagues demonstrated, in 2015, that Nek5 is located in the mitochondria, where it interacts with local proteins and alters ROS production and energy rates [16] (Figure 2). The knockdown of Nek5 increased basal and complex IV oxygen consumption and increased ROS generation (Figure 3). The increase in ROS levels in Nek5 knockdown cells is likely to be related to an increase in OXPHOS activity and cell respiration. In contrast, during Nek5 overexpression, oxygen consumption in complex IV and complex II was significantly reduced, although no change in basal respiration was observed. Accordingly, ROS levels decreased in Nek5 overexpression cells [16].

Pavan and colleagues recently showed that Nek6 regulates ROS levels in prostate cancer Du145 cells [52]. Nek6 knockout increases intracellular ROS and decreases antioxidant enzyme expression, such as SOD1, SOD2, and PRDX3). These cells also showed a decrease in mitochondrial membrane potential [52]. Finally, Riback and colleagues observed that several important metabolic enzymes have their expression levels and activity altered [21] in the Du145 Nek6 KO cells. Specifically, it was found that the pyruvate dehydrogenase (PDH) expression level is decreased in Nek6 knockout cells and that the levels of both LDHA and PFK1 are increased in Nek6 knockout cells (Figure 3). Together, these data suggest that the levels of lactate production and glycolysis are increased, whereas the production of acetyl-CoA is diminished in the knockout cells [21]. These observations can explain why the Nek6 knockout cells show a lower cellular respiration level.

### 3.2. Mitochondrial Respiration and Metabolism Partners

Among Nek4 interactors, we found several proteins related to OXPHOS, such as ATP5F1 subunits, NDUFS1, and UQCRC2 [13]. In the Nek10 interactome, we also found NDUFS1, ATP5J, and UQCR10 [19] (Figure 1 and Figure 3). Because we observed that Nek4 does not enter the mitochondria [20], we believe that these interactions can also be indirect through complexes of proteins associated with mitochondrial membranes. Regarding Nek10, although we are not sure if Nek10 enters the mitochondria, or at least which isoform does so, we cannot exclude the possibility of a direct interaction.

Nek5 was shown to interact with COX11, an intrinsic mitochondrial membrane protein essential for the assembly of an active cytochrome *c* oxidase complex (complex IV) in a yeast two-hybrid screening [16]. In 2022, an analysis of Nek5-overexpressing MCF10A cells proteome using BioID revealed interesting mitochondrial partners such as HIGD1A, TFAM, C2orf47, SLC25A12, GSR, TIGAR, IARS2, IDH3B, CLPX, TFB2M, MFN2, PPA2, MRPS11, and MT-ATP6 (from the highest to the lowest logFC, *p*-value < 0.05) [18]. The interactome enrichment pathway analysis of NEK5 interactors showed that mitochondrial organization and mitochondrial gene expression are among the top five enriched pathways in MCF10A Nek5-overexpressing cells, reinforcing our hypothesis of Nek5 participation in mitochondrial mtDNA homeostasis and mitochondrial functions (Figure 1 and Figure 3).

Finally, in addition to interactors from oxidative phosphorylation, several proteins from the glycolytic pathway were found in the Nek family interactome. Phosphofructokinase (PFK1) and pyruvate dehydrogenase E1 component subunit beta (PDHB) were found in the Nek4 and Nek10 interactomes, respectively [13,19]. Moreover, pyruvate kinase M (PKM) was found in both interactomes of Nek1 and Nek4 [13,22]. PKM is a key enzyme for glycolysis, and the PKM2 isoform is one of the main players in switching between oxidative and glycolytic metabolism associated with the Warburg effect in cancer cells (for review, see [53]).

In addition to the switch from OXPHOS to glycolysis, some tumor cells increase the degradation of fatty acids, thus increasing the ATP production via beta-oxidation or supply metabolic intermediates, such as acetyl-CoA or citrate (for review, see [54]). The upregulation of enzymes from the metabolism of lipids has been described in several cancers [55].

The citrate synthase (CS) is an enzyme present in almost all living organisms [56] and is located in the mitochondrial matrix [57]. CS participates in the TCA cycle [58], which catalyzes the formation of citrate from acetyl-CoA and oxaloacetate [57]. The control of the citrate synthase activity occurs through its substrates and products. Citrate inhibits CS, while oxaloacetate binding increases the affinity of CS for acetyl-CoA [57]. Among all interactors, Peres de Oliveira and colleagues (2020) showed that CS could interact with Nek10. In addition, they confirmed the interaction between Nek10 and CS by proximity ligation assay (PLA) [19] (Figure 1 and Figure 3). Also, the citrate synthase activity is decreased in Nek10-depleted HeLa cells [19]. The decrease in the enzyme citrate synthase activity is related to H55 polymorphism [59] and lipotoxicity [60]. Citrate synthase activity was also decreased in Nek5-K33A (kinase dead)-overexpressing cells, compared to both NEK5-WT and control cells, suggesting a potential kinase activity dependency of Nek5 on mitochondrial metabolism [17]. Interestingly, TOM20 protein levels were higher in Nek5-kinase dead cells while remaining unchanged in Nek5-WT overexpressing cells. Nek5-kinase dead cells also showed an increase in mitochondrial membrane potential (MMP) [17]. The increase in TOM20 levels and MMP in Nek5-K33A overexpressing cells might indicate compensatory signaling in order to increase energy production in response to the decreased CS activity, suggesting that Nek5 kinase activity may be necessary for the maintenance of essential mitochondrial functions.

The 2,4-dienoyl-CoA reductase (DECR1), a rate-limiting enzyme that participates in beta-oxidation, was found in the Nek1 interactome [22]. DECR1 has already been implicated in fatty acid oxidation (FAO) that is important to sustain cancer cell survival [61,62]. Glycogen synthase kinase 3 (GSK3) phosphorylates and inactivates glycogen synthase and is a metabolic sensor, integrating protein and lipid synthesis and glucose and mitochondrial metabolism. In mitochondrial metabolism, GSK3 decreases oxygen consumption and inhibits mitochondrial biogenesis by peroxisome proliferator-activated receptor gamma coactivator 1-alpha (PGC1A) inhibition [63]. Also, GSK3 induces mitochondrial fission via DRP1 phosphorylation [64]. GSK3 also phosphorylates the ATP-citrate lyase (ACLY), inhibiting the fatty acid biosynthesis [65]. ACLY is a metabolic enzyme that catalyzes the breakdown of citrate to generate acetyl-CoA, a common substrate for de novo cholesterol and fatty acid synthesis [65]. ACLY was found in the Nek5 interactome (Figure 1).

The Nek10 interactome also shows proteins related to the glycolytic pathway and TCA interface, including the pyruvate dehydrogenase (PDHB) that catalyzes the overall conversion of pyruvate to acetyl-CoA and CO_2_. Also, glutamate dehydrogenase (GLUD1) is a Nek10 interactor. GLUD1 is localized in the mitochondrial matrix and catalyzes the reaction of L-glutamate into alpha-ketoglutarate [66], contributing to both Krebs cycle-driven anaplerosis and energy production. PLA (Proximity Ligation Assay) and confocal immunofluorescence microscopy were used to test the interaction of Nek10 and GLUD1. Both analyses showed the co-localization of GLUD1 and NEK10 [19].

According to results showing the role of Nek6 in ROS level regulation [52], the yeast two-hybrid screening performed by Meirelles and colleagues identified the peroxidase PDRX3 as a Nek6 interactor as well as the regulatory protein VWA8 (Von Willebrand A Domain Containing Protein 8) [23], which is an AAA+ ATPase that is localized to the mitochondrial matrix face of the inner mitochondrial membrane, and its depletion is associated with increased oxidative capacity [67].

## 4. Neks and Mitochondrial Morphology

Mitochondria are dynamic organelles, and when visualized in mammalian cells, they can be presented as a filamentous network or small round shapes. The changes in the mitochondrial morphology or connectivity are essential for responding to cellular conditions. Mitochondrial homeostasis relies on the opening and closing of permeability pores and the mitochondrial fission and fusion processes (for review, see [68,69,70]). Mitochondrial fusion requires Mitofusin 1 and 2 (Mfn1 and Mfn2) in the outer mitochondrial membrane and Opa1 in the inner mitochondrial membrane to assemble elongated mitochondrial networks, which can favor ATP synthesis and decrease mitophagy of the mitochondria with elevated mitochondrial membrane potential (MMP) [71].

On the other hand, mitochondrial fission requires dynamin-related protein 1 (Drp1), which is needed to correct for the segregation of mitochondria for cell division, mitochondrial biogenesis, and mitophagy of damaged mitochondria [71,72,73]. Mitochondrial fission factor (MFF) is the main Drp1 receptor in mammalian mitochondria [74].

Depending on cell type, one cell can present a mixture of mitochondria with different morphologies from long filaments to small round mitochondria. The equilibrium of these different forms is important for the proper maintenance of mitochondrial functions. Changes in the cytoskeleton and contacts with organelles such as endoplasmic reticulum, lipid droplets, and Golgi, in addition to mitochondrial dynamic protein defects, can change the morphology of the mitochondria population (for review, see [68]).

The internal structure of the mitochondria, determined by the MIM fold, sustains the components of ETC and preserves the content of the mitochondrial matrix. OPA1 is the main player in cristae structure maintenance [75,76]. Moreover, the lipid composition of the mitochondrial membranes is essential to allow the fusion/fission events as well as the cristae modulation [77].

### 4.1. Nek1, Nek6, and Nek10 Depletion Are Associated with Mitochondrial Fragmentation, as Opposed to Nek4

Wang et al. (2021) observed abnormal mitochondrial fragmentation in Nek1^Kat2J/Kat2J^ mouse embryonic fibroblasts by confocal imaging using MitoTracker [44]. The normal phenotype was rescued by treatment with acetyl-L-carnitine (ALCAR), which facilitates the transfer of fatty acids from the cytosol to the mitochondria [44].

Moreover, Nek1 has also been shown to mediate neural development [44]. Nek1 loss-of-function mutants confer susceptibility to amyotrophic lateral sclerosis (ALS) [78,79]. Abnormal mitochondrial structure and function in motor neurons have been implicated in the neurodegenerative process of the disease [80,81]. It remains elusive whether ALS-related Nek1 mutants would also impair mitochondrial morphology, causing neurological dysfunction. In addition, since mitochondria can replicate by fission in axons [82], a putative link between mitochondrial permeability regulation and the mitochondrial fission process in motor neuron cells, with a possible role for Nek1 as a regulator, is also attractive but needs to be further investigated.

Furthermore, mitochondrial homeostasis is also dependent on normal retromer function. Nek1 regulates retromer-mediated endosomal trafficking by phosphorylating a component of the retromer complex known as vacuolar protein sorting subcomplex 26 (VPS26B) at residues S302/S304 [44]. VPS26B phosphorylation by Nek1 prevents the binding of sorting nexin-27 (SNX27), a major regulator of endosome-to-plasma membrane recycling of proteins [44].

VPS26B knockdown in mouse embryonic fibroblasts also led to mitochondria morphology defects [44]. Wang et al. (2021) showed that normal mitochondria morphology is rescued when wild-type, but not phosphodeficient S302A/S304A double mutant VPS26B, is expressed in VPS26B knockdown cells [44]. Thus, since retromer mediates cellular trafficking such as the endosomal transport of receptors, retromer dysfunction due to Nek1 deficiency leads to pleiotropic defects at the cellular level, including mitochondrial structural abnormality and dysfunction [44]. The retromer complex also plays a critical role in the development of the nervous system [83]. It remains unclear whether VPS26B phosphorylation by Nek1 is a valid regulation checkpoint in normal neural development.

Although the effects of Nek1 deficiency in mitochondrial morphology are beginning to be elucidated [44], the molecular mechanisms underlying the role of Nek1 in maintaining normal mitochondrial ultrastructure are still unclear. Also, little is known about the consequence of ALS-related mutant expression in mitochondrial morphology and function and its relation to neural development. One possibility could be the mitochondrial morphologic changes observed in Nek1 deficiency secondary to defects in VDAC phosphorylation, which would lead to proton leakage and mitochondrial membrane potential [45].

Nek10 deficiency also promotes a severe effect on mitochondrial morphology [24] (Figure 2 and Figure 4). Nek10-depleted HeLa cells presented more fragmented than elongated mitochondria compared to the control cells [19]. It is known that the process of fusion and fission coordinates mitochondrial morphology, which is intrinsically associated with mitochondrial bioenergetics, but the mechanism related to this morphologic change needs to be further investigated (Figure 4).

For the Du145 Nek6 knockout cells, Riback and collaborators also found a clear mitochondrial fragmentation phenotype (Figure 2 and Figure 4, [21]), although further studies are required to unravel the mechanisms and key molecules involved.

Through immunofluorescence analysis and Western blotting of fractionated cells, we observed that Nek4, which was partially co-distributed with mitochondria, was abundantly present in mitochondrion-associated membranes [20]. A change in mitochondrial morphology was observed in correlation with Nek4 expression levels. Nek4 knockdown cells showed longer mitochondrial filaments and reduced cristae numbers, while Nek4 overexpression resulted in increased mitochondrial sphericity and decreased branch length [20] (Figure 2 and Figure 4). Based on these findings, the expression of proteins related to mitochondrial fission and fusion was analyzed. It was observed that Nek4 increases DPR1 phosphorylation, which induces mitochondrial fragmentation when phosphorylated. As for mitochondrial fission, Nek4 overexpression resulted in FIS1 and MFF upregulation, while Nek4 knockdown resulted in FIS1 downregulation [20].

Nek4 interaction with DRP1 was further investigated through a proximity ligand assay, which revealed that Nek4 and DRP1 are close enough to interact or form a protein complex (Figure 4). These interactions are potentiated with Nek4 overexpression [20]. It was also found that Nek4 overexpression activates Erk1/2, one of the main kinases responsible for DPR1 phosphorylation. Finally, to investigate whether DRP1 increased phosphorylation is linked to the increase in mitochondrial respiration with a higher Nek4 expression, a DPR1 inhibitor, Mdivi-1, was used, and it blocked the Nek4 effect on basal and ATP-coupled respiration, leading to the conclusion that Nek4 participates in Erk1/2-mediated Drp1 activation, resulting in mitochondrial fission and increased respiration [20].

It is interesting that Nek4 overexpression is associated with mitochondrial fission as well as an increase in mitochondrial respiration and protection against ROS. An increase in mitochondrial fission is usually associated with a decrease in mitochondrial functions (respiration and quality); however, in several tissues, it has already been demonstrated that fission induction is necessary to handle the excess of nutrients and low ATP demand [84,85]. For example, in β-cells and brown adipocytes or neuronal synapses, mitochondrial fission is required for proper function [86,87,88,89]. Moreover, fission is important for mitochondrial quality control (MQC), which removes damaged mitochondria. Additionally, it is important to notice that fission is critical for mitochondrial biogenesis [90] and the correct mitochondrial segregation during the cell cycle [73]. Thus, one might conclude that Nek4 increases mitochondrial respiration by increasing fission and the quality of the mitochondria. On the other hand, an excess in mitochondrial fission is also associated with cancer invasion and chemoresistance [91,92].

### 4.2. Nek Interactors Related to Mitochondrial Morphology

We lack information on the effect of Nek5 in mitochondrial morphology; however, among its interactors found in yeast two-hybrid screening [16], Metaxin 2 (MTX2) was identified. In a screening to identify mitochondrial regulators of the type I interferon response that is activated by mtDNA detection in the cytosol, He and colleagues (2022) have found MTX2 to be essential for maintaining the cristae architecture [93].

Moreover, Mitofusin-2 was detected in the NEK5 BioID screening [18], suggesting that, although still not investigated, NEK5 and other members of the Nek family might affect the mitochondrial morphology.

The ATPase family AAA domain-containing protein 3 (ATAD3) proteins (ATAD3A and ATAD3B) are mitochondrial membrane-bound ATPases [94]. The ATAD3 is localized to the mitochondrial inner membrane, and ATAD3A is implicated in cristae maintenance [95,96]. They are essential for the regulation of mitochondrion–ER interactions and mitochondrial biogenesis processes. In humans, mutations in ATAD3A cause severe neurological syndromes [97]. In addition, through the negative regulation of ATAD3A function, ATAD3B supports mitochondrial stemness properties [97]. ATDAD3A was found to be a potential partner in Nek10 interactome [19] (Figure 1), suggesting that, along with the alteration in mitochondrial morphology in Nek10-depleted cells, Nek10 might also be involved in the regulation and signaling of mitochondrial morphology.

The effect of Nek4 on mitochondrial morphology could result from its interaction and regulation of DRP1 since proximity ligand assays showed close Nek4–DRP1 proximity, although the direct interaction was not observed. Moreover, the phosphorylation of DRP1 at S616 is increased by Nek4 overexpression as well as the activation of ERK1/2 [20], one of the most known activators of fission by the phosphorylation of DRP (Figure 4). Mfn1 was found in the Nek4 interactome [13], and ongoing studies confirmed this interaction and aim to clarify the mechanism.

Mitochondrial import inner membrane translocase TIM14 (DNAJC19 gene) was identified in the Nek6 interactome [23]. The function is not completely understood, but due to its MIM localization, it can be involved in protein and cardiolipin transport [98]. Mutations in the DNAJC19 gene led to dilated cardiomyopathy with ataxia (DCMA), a condition in which mitochondrial fragmentation was observed [99].

## 5. mtDNA Maintenance

Mitochondria have their own DNA, a circular, double-strand molecule of about 16.5 kb that encodes 13 essential oxidative phosphorylation (OXPHOS) components, in addition to rRNAs and tRNAs [100]. The mtDNA is concentrated in structures named nucleoids, which are enriched in mitochondrial transcription factor A (TFAM) responsible for mtDNA compaction or relaxing, regulating the accessibility for replication, transcription, or repair machinery [101,102]. The replication of the mtDNA is performed by DNA polymerase gamma (POLG), twinkle helicase, and single-stranded DNA binding protein (SSBP1) activities (for review, see [103]).

The mtDNA is highly susceptible to oxidative damage, mainly by chronic exposure to the ROS generated by OXPHOS reactions, complicated by the lack of histones and mitochondrial DDR-specialized enzymes [104,105]. The base excision repair (BER) is the predominant repair pathway in mitochondria and the first repair pathway characterized in the mitochondria (for review, see [106]). BER is essential for the repair of mtDNA damage caused by ROS and involves three main steps: recognition and removal of the oxidized base, end processing of the apurinic/apyrimidinic (AP) site, and gap filling. The most prominent enzymes for each step are DNA glycosylase (OGG1), AP endonuclease 1 (APE1), and LIG3 and flap endonuclease (FEN1 gene), respectively. The genetic ablation of any gene of BER enzymes causes mouse embryonic lethality (for review, see [106]). Currently, the existence of other repair pathways in mammalian mitochondria is still controversial. Enzymes related to mismatch repair (MMR) and nucleotide excision repair (NER) have already been found in mammalian mitochondria extracts, and their ablation causes mitochondrial defects [107,108,109,110]. The mitochondrial double-strand break repair (DSBr) is even more contradictory because the double-strand damage would originate from a linear DNA fragment that would be further degraded. Moretton and colleagues (2017) observed no evidence of DSB repair and a reduction in mtDNA content after the induction of mtDSB [111]. However, the presence of some specific isoforms of enzymes related to DSBr was documented in mitochondria [107,112,113], and some evidence of the occurrence of homologous recombination (HR) in this organelle was presented [114]. It is interesting to note that most of the enzymes that participate in mitochondrial DNA damage response are splicing variants of the nuclear DDR pathways [115].

Depending on the cell type, hundreds to thousands of mtDNA copies can be present in the cell. The mtDNA content is regulated by transcription and is directly related to TFAM levels (for review, see [116]) [117,118] and affected by the metabolic demand of the cell. Although changes in mtDNA copy number are usually associated with a compensatory mechanism in mitochondrial diseases and a decrease in the severity of cancer, aging, and neurodegenerative disorders, the link between mtDNA copy number and disease severity is not straightforward [119].

Reduced mtDNA copy number in peripheral blood leukocytes enhances the risk of coronary heart disease [120]. Also, the increased mtDNA content is associated with a lower incidence of chronic kidney disease [121]. In addition, mtDNA content in peripheral blood is associated with sudden cardiac death risk [122]. However, the mtDNA content is increased in attention-deficit hyperactivity disorder.

### 5.1. Nek Involvement in mtDNA Maintenance

Nek1 knockout cells showed an increase in mitochondrial mass, evaluated by Mitotracker green, which is associated with an increase in the mtDNA content [46]. This mtDNA showed poor integrity and was more susceptible to damage by methyl methanesulfonate (MMS) (Figure 5). The increase in mtDNA damage observed in Nek1 absence can be related to a loss of repair capacity, and indeed, RNAseq results demonstrate reduced expression of XPA and XPC in Nek1 KO cells [46] (Figure 5). These enzymes are important for the recognition of DNA damage and nucleotide excision repair (NER). On the other hand, we observed an upregulation of several base excision repair (BER) genes, such as APEX1, APEX2, POLκ, PCNA, and FEN1 [46]. The increase in the expression of these genes could be a compensatory mechanism to protect the mtDNA integrity. Lig3 is downregulated in this condition [46]. The MSH2 and PMS2 genes are also upregulated in Nek1 KO cells [46].

Although, so far, Nek1 was not implicated directly in the maintenance of mtDNA, several studies implicate Nek1 in the maintenance and repair of nuclear DNA [123,124]. In this context, Nek1 co-localized with γ-H2AX and NFBD1/MDC1 [123]. Furthermore, studies from Liu and colleagues (2013) showed that Nek1 kinase activity stabilizes the complex between the checkpoint kinase ATR (ATM and Rad3-related) and its partner ATRIP (ATR-interacting protein), which is important for efficient DNA damage signaling [124].

Nek4 has already been implicated in nuclear DSB repair through the recruitment or stabilization of the complex DNA-PK (DNA-PKcs-Ku70-Ku80) since Nek4-depleted cells showed reduced p53 activation and H2AX phosphorylation after DNA damage [125].

The mtDNA integrity was found to increase in Nek4 overexpressing cells, and the Nek4 knockdown, in turn, led to a decrease in mtDNA integrity, and in both conditions, the mtDNA content was unchanged [20] (Figure 5). Fen1 and APE were proteins found in the Nek4 interactome, and indeed, Fen1 expression levels were increased in Nek4 overexpressing cells [13,20] (Figure 1). Our results suggest a Nek4 role in protecting the mtDNA from damage by increasing antioxidant defenses or increasing its repair through BER [20] (Figure 5).

Our research group has previously reported the involvement of Nek5 in DDR through topoisomerase IIβ dynamic interaction [126]. Topoisomerase Iiβ (TOP2B) is the only known type II topoisomerase present in mitochondria, although studies suggest that there might be other active enzymes that catalyze the separation of intertwined mtDNA molecules [127]. Due to Nek5 localization in mitochondria, its participation in nuclear DDR, and its interaction with Topoisomerase IIβ, we decided to investigate whether Nek5 could also participate in mtDNA maintenance.

Nek5 was demonstrated to interact with LonP1 and later on, with TFAM [17,18], a key protein for mtDNA packaging, remodeling, and transcription [128]. We demonstrated that cells overexpressing Nek5 showed an increase in mtDNA integrity, with a decrease in mtDNA copy number, while the overexpression of Nek5^K33A^, a kinase-dead version, showed a significant decrease in the mtDNA integrity, with no change in mtDNA copy number (Figure 2). Upon mtDNA oxidative damage, the mtDNA of cells expressing Nek5^WT^ was demonstrated to be more resistant to zeocin and rotenone treatment when compared to both control and Nek5^K33A^, suggesting a kinase-dependent activity in mtDNA maintenance and possibly, in mtDNA repair [17]. Moreover, Nek5-LonP1 interaction increased upon oxidative damage, suggesting that this interaction might be related to mtDNA Damage Response events [17]. Since BER is the predominant mechanism for mtDNA repair [129], we evaluated the expression of key mitochondrial BER enzymes and enzymes related to mitochondrial replication and biogenesis. We wanted to investigate whether Nek5 was involved in mtDNA repair or mitochondrial biogenesis/turnover as we also observed an increase in mtDNA copy number upon Nek5^WT^ expression.

The levels of TFAM and PGC1A were not significantly altered, suggesting that Nek5 might not be involved in mitochondria biogenesis. On the other hand, both POLG, the only DNA polymerase in mitochondria [130], and APE1 (AP endonuclease 1) involved in the mitochondrial BER pathway [121], were downregulated in Nek5^K33A^ cells in variance with Nek5^WT^ cells, where those genes were upregulated [17]. This could be related to a potential Nek5 participation in mtDNA repair when both proteins are crucial for mitochondrial base excision repair (mtBER).

Recently, our group identified TFAM, TFBM, and CLPX, proteins associated with mtDNA maintenance, transcription, and repair [131,132,133,134], as potential direct interaction partners in a BioID screening [18], reinforcing our hypothesis of Nek5 participation in mtDNA biology. Nek5 participation in the LonP1-TFAM axis, which is related to mtDNA transcription and remodeling [132,134], the alteration in the expression of BER genes, the kinase-dependent involvement of Nek5 with mtDNA integrity, and mtDNA interaction proteins strongly suggest the participation of Nek5 in the mtDNA damage response, although the mechanisms underlying these functions are still unknown.

We recently described that, in Nek10-depleted HeLa cells, the mtDNA content was increased compared to control cells [19]. mtDNA integrity was also affected in Nek10-depleted HeLa cells (Figure 2 and Figure 5). Moreover, the depletion of Nek10 increased the damage of mtDNA upon zeocin treatment (Figure 5). Also, the mass of mitochondrial proteins was accessed in Nek10-depleted HeLa cells, TOM20, TFAM, VDAC, and OXPHOS complex II; all those proteins showed low expression compared to control cells [19].

Although the direct effect on mtDNA integrity is not known so far, Nek6 knockout in Du145 cells is associated with an increase in ROS levels and on the other hand, a decrease in the levels of antioxidants enzymes, SOD1, SOD2, and PRDX3 [52]. The last one was also identified in the Y2H screening of Nek6 interactors [23] (Figure 1 and Figure 5).

### 5.2. Neks and the Interaction with DNA Repair-Related Enzymes

It is interesting to highlight the relationship between mtDNA and mitochondrial cristae. The mtDNA nucleoid is physically associated with IMM (for review, see [135]), and proteins involved in the cristae maintenance through MICOS assembly (mitochondrial contact site and cristae organizing system), such as Mic60/Mitofilin, have already been demonstrated to be essential for the nucleoprotein complex organization [136]. The MTX2 and ATAD3A, interactors of Nek5 and Nek10, respectively, have already been found to be important for cristae maintenance. The depletion of MTX2 or ATAD3A leads to cristae disruption and cytosolic mtDNA release and inflammation by IFN-I response activation [93,137].

As mentioned before, both Nek1 and Nek4 depletion cause cristae loss, which could be associated with mtDNA damage. The information on the internal structure of mitochondria after Nek10 or Nek5 depletion is still lacking, and it would be interesting to verify cristae conditions in these cells. In addition to cristae maintenance, a direct effect on repair pathways could justify the changes in the integrity of mtDNA. Regarding interactors that have not been confirmed so far, SSBP1 was found in the Nek4 interactome. SSBP1, in addition to its importance to mtDNA replication through the stabilization of the replication fork, is also important for mtDNA integrity because it protects the unpaired strands from degradation during the replication or repair (for review, see [138]).

Interestingly, both Nek4 and Nek5 were found to co-immunoprecipitate with (IP-LC/MS-MS) PARP1 [13,16] (Figure 1). PARP1 is also located in the mitochondrion and is a sensor of single-strand breaks. It is possible that it participates in mtDNA repair (for review, see [139]).

Among Nek10 interactors, we have found that DNA-(apurinic or apyrimidinic site) endonuclease (APEX1 or APE1), an enzyme with multiple functions, plays a role in DNA repair and reduction/oxidation signaling. APEX also participates in the BER pathway [140], which occurs in AP site cleavage [141]. This protein can be found in the nucleus, cytoplasm, and mitochondria [142] (Figure 5). PARP and APEX also participate in nuclear DNA damage, and the interaction with these enzymes is not an indication of direct interaction inside mitochondria.

**Figure 5 cells-13-00473-f005:**
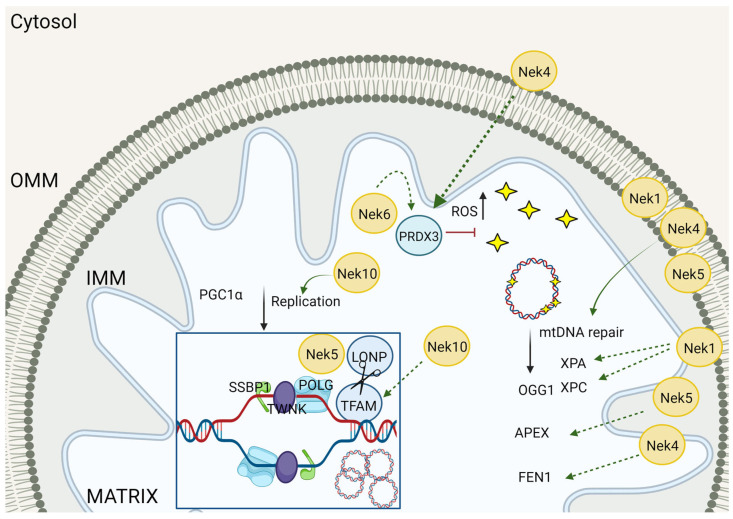
Neks maintain mtDNA integrity. The mtDNA content is regulated by the transcription wherein POLG is the main polymerase, TFAM is a transcription factor that composes the mtDNA nucleoid and controls the transcription by allowing access to mtDNA, and LonP1 is a protease that also controls replication by the degradation of mtDNA core proteins, which include TFAM [128,133,143]. Nek4 increases mtDNA integrity with no changes in mtDNA content, possibly increasing the transcription of DNA repair and antioxidant enzymes. Arrows indicate activation, interaction, or functional regulatory interference. Dashed arrows indicate changes in protein or mRNA expression. Red lines indicate inhibitory regulatory influence. Light blue circles represent putative or confirmed interactors of the Neks. Created with BioRender.com (accessed on 23 January 2024).

An increase in mtDNA content and damage was observed in Nek1-depleted cells, while an increase in the base excision repair (BER) gene expression has been observed. The same trend was observed in the case of Nek10 depletion, which showed an increase in mitochondrial mtDNA with low quality. Also, a mitochondrial mass decreased in those cells, indicating that the highly fragmented mitochondria are probably losing the mtDNA content. Nek5 overexpression also changes mtDNA content, increasing it and protecting it from damage. The increase in POLG expression has been observed in these cells and the APEX. Nek5 also can interact with LONP and TFAM, important regulators of mtDNA replication. The direct effect of Nek6 in mtDNA is unknown so far, but Nek6 KO cells show high levels of mitochondrial ROS, and PRDX3 is a putative interactor of Nek6.

## 6. Response to Stress and Cell Death

Mitochondria play a central role in both cellular redox metabolism and programmed cell death (PCD) induction since they contain certain agents relevant to these two key cellular functions. Both processes have been vastly reviewed in the recent literature [144,145,146,147]. Briefly, apoptosis is a death process closely related to mitochondria. For intrinsic apoptosis occurrence, the permeabilization of the mitochondrial outer membrane should occur. Basically, upon several types of stimuli such as nuclear DNA damage, mitotic arrest, or growth factor withdrawal, the pro-apoptotic BCL-2 proteins BAX and BAK form porous in the mitochondrial membrane, leading to mitochondrial outer membrane permeabilization (MOMP) and release of mitochondrial content to the cytosol. The main activator of apoptosis is the cytochrome C, which activates apoptotic protease-activating factor 1 (APAF1) and thus recruits caspases [147].

Regarding the extrinsic apoptosis pathway, a common activation is via death receptors, such as Tumor necrosis factor receptor (TNFR) or Fas, whose activation by Fas-ligand TNF alpha, TNFRS10, or TRAIL (tumor necrosis factor ligand) triggers the activation of caspase-8, and then -3, and -7. Both types of apoptosis are connected by the cleavage of BID (BH3-interacting domain death agonist) that also induces MOMP (for review, see [147]). In addition to apoptosis, mitochondria can be associated with necroptosis cell death. This form of death is also associated with TNFR activation, which, in the case of caspase inactivation, leads to the phosphorylation of the receptor-interacting protein kinase 1 (RIPK1) and RIPK3 and necrosome formation and thus, plasma membrane permeabilization. RIPK3 also activates the mitochondrial pyruvate dehydrogenase complex (PDH), which increases OXPHOS and ROS generation, which consequently exacerbates necrosome formation [148].

The cell fate between cell death and survival depends on MOMP levels that are considered the no-return point for cell death. In case of extensive damage, the MOMP of a subpopulation of mitochondria would lead to the cleavage of BID that would propagate the MOMP of other mitochondria [149]. However, cells can maintain ATP production even after cytochrome C release [150]. Important for the glycolysis during MOMP is the GAPDH enzyme, which allows the autophagy activation with elimination of depolarized mitochondria and cell survival [151]. Survival is associated with an inhibition of caspase activity and cleavage of mitochondrial products released to the cytosol. The regulation of this decision is particularly important in the study of cancer cell response to treatment since most anticancer agents work by activating intrinsic apoptosis. However, a relationship with the increase in the expression of FAS or TRAIL receptors, or inhibition of RIPK1 or RIPK3 has also already been associated with cancer cell sensitivity to anticancer agents (for review, see [144]).

The metabolite flux in the outer mitochondrial membrane is mainly controlled by the voltage-dependent anion-selective channel (VDAC) protein [69]. Seo et al. (2019) recently showed that DRP1 receptors and MFF isoforms 1 and 2 directly interact with VDAC1 and that the MFF-VDAC1 complex is an important regulator of mitochondrial cell death in cancer cells [152]. However, a mechanistic understanding of this regulation is still lacking.

### Nek1 and Nek5 Participate in Cell Death in Opposite Ways

Nek1 was previously shown to phosphorylate VDAC1 at S193 and consequently regulate the pore opening/closing and outer membrane permeability [14,43] (Figure 6). In Nek1’s absence, VDAC1 is not phosphorylated properly, allowing easy loss of mitochondrial membrane potential, leakage of cytochrome C, and cell death. In fact, Nek1^Kat2J/Kat2J^ mice lose VDAC1 phosphorylation under genotoxic treatment [14]. Increased expression of Nek1 promotes persistent phosphorylation of VDAC1 in renal carcinoma cells, preventing apoptosis under DNA-damaging agent treatment [153]. Nek1-deficient cells are characterized by increased mitochondrial membrane permeability and accelerated cell death [14,15]. Singh and co-workers found Nek1 as an interactor of TLK1, a tousled-like kinase, important for DNA replication, transcription, and repair. Moreover, they demonstrated that Nek1 is a TLK1 substrate and that Nek1 phosphorylation at T141 by TLK1 contributes to the interruption of the cell cycle and activation of the DDR to mediate survival after damage [43]. In 2020, the same group observed that the overexpression of Nek1 hypoactive mutant (T141A) decreases mitochondrial respiration in tumor cells, but not in HSG cells. The authors suggested that one possible explanation could be that cancer cells could depend on the axis TLK1/Nek1/VDAC, avoiding mitochondrial permeability and cell death. According to the authors, after extensive DNA damage, the phosphorylation of Nek1 by TLK1 at T141 and the recruitment of Nek1 to the nucleus prevents VDAC phosphorylation by Nek1, which, in turn, allows the opening of VDAC, leading to mitochondrial permeability leakage, reduced mitochondrial respiration, and activation of the intrinsic apoptotic pathway [43]. The relationship between death induced by DNA damage was also observed by Pelegrini and colleagues. They showed that Nek1-stable knockdown presents a delay in DNA repair after different types of damage. Mainly, after cisplatin treatment, they observed reduced activation of Chk1 and increased sensitivity to the treatment compared to control cells [154]. The same deficiency in DNA repair after cisplatin-induced DNA damage in Nek1 knockdown cells was also observed by our group, and according to our interactome data, we could conclude that NEK1 participates in DDR for correcting DNA crosslinks [22]. These studies indicate a sensitivity to death in the case of Nek1 depletion related to DDR deficiency, but not directly related to a PCD activation.

Nek1-deficiency in nek1^Kat2J/Kat2J^ mice promotes the activation of receptor-interacting serine/threonine-protein kinase 1 (RIPK1), necroptosis, and apoptosis in cerebrovascular endothelial cells [44]. In fact, Nek1 binds to activated RIPK1 and restricts RIPK1-dependent apoptosis (RDA) by negatively regulating the formation of a highly ubiquitinated and activated RIPK1 pool [44] (Figure 6).

The regulation of cell metabolism by Nek5 kinase is also likely to have implications for cell fate. Hanchuck and colleagues [16] have demonstrated that, upon hydrogen peroxide treatment, Nek5 knockdown cells show an increase in cell death, which correlates with the increase in ROS production in this model (Figure 2). In variance with these results, cells overexpressing Nek5 show an increased resilience to hydrogen peroxide treatment, evidenced by the increase in survival, which also correlates with decreased ROS levels in those cells [16] (Figure 6).

The relationship between Nek5 and apoptosis was also explored in muscle cells, where Nek5 was found to be a caspase-3 cleavage substrate [156]. In variance with the previous findings, upon Nek5 overexpression, the percentage of apoptotic cells was increased, followed by an increase in caspase-3 activity. These results suggest that the role of Nek5 in apoptosis might be indirect and context- and cell-dependent and requires further exploration (Figure 6).

Nek4 does not seem to be involved in caspase-mediated cell death. Park and colleagues (2017) investigated TRAIL (apoptosis-inducing ligand) resistance in four lung cancer cell lines and observed that Nek4 depletion increases the sensitivity to cell death induced by TRAIL. They tried different cell death stimuli, such as etoposide, which activates the intrinsic apoptotic pathway, TNF-α/cyclohexamide, which activates the extrinsic pathway, and TRAIL, with the pan-caspase inhibitor zVAD. The results indicated that Nek4 is only involved in the regulation of the TRAIL-mediated cell death pathway [157].

The mechanism responsible for TRAIL sensitization by Nek4 depletion was investigated through the analysis of the expression of various targets of TRAIL resistance, such as TRAIL receptors and anti-apoptotic and pro-apoptotic proteins. They found that the anti-apoptotic protein survival (BIRC5) decreased significantly in response to the inhibition of Nek4 in TRAIL-treated cells [157] (Figure 6). It is interesting to note that Nek4 depletion also sensitizes cells to death caused by vincristine, but this effect does not seem to be related to mitochondria and is more related to microtubule polymerization function since the treatment with paclitaxel induces minor mortality in these cells [158]. On the other hand, Nek4 knockdown cells are resistant to DNA damage induced by double-strand break agents, such as ionizing radiation or etoposide. Again, no direct relation with apoptosis control was presented, and the observed effect was attributed to a deficiency in the activation of DDR [125].

A clear link between Nek10 and cell death has not yet been found. Moniz and colleagues showed that Nek10 participates in the maintenance of the G2/M checkpoint followed by ultraviolet (UV) irradiation [8]. Also, after DNA damage insult, such as cisplatin treatment, the loss of Nek10 sensitizes cells, probably due to a deficiency in DDR activation by p53. NEK10 kinase phosphorylates p53 at Y327 (tyrosine 327), inducing the transcription of several genes such as p21 [37]. The phosphorylation of p53 by Nek10 indicates an interesting link with the role of Nek10 in mitochondria. As mentioned before, Nek10 depletion is associated with proton leak, and in the literature, it was observed that the basal proton leak could act in cell protection [159] and p53 regulates mitochondrial respiration by performing a balance between the respiration and glycolysis pathways [160]. So, Nek10 also could be important for the regulation of mitochondrial function through p53 phosphorylation.

The apoptosis-inducing factor (AIF) was found in the Nek4 and Nek10 interactomes (Figure 1). AIF is found in the mitochondrial intermembrane space (IMS), and it is released from the IMS to the cytosol in response to apoptotic stimuli and then translocated into the nucleus, where it acts as a proapoptotic factor [161] (Figure 6). Moreover, poly(ADP-ribose) polymerase-1-(PARP1), found in Nek4 and Nek5 interactomes (Figure 1), is necessary for AIF nuclear translocation [162].

## 7. Response to Stress and Autophagy

As mentioned before, the mtDNA’s genes encode 37 proteins or RNAs, including some OXPHOS proteins and rRNAs or tRNAs. The other mitochondrial proteins are nuclear-encoded and transcripted at the cytosol and then enter the mitochondria through membrane transporters (TOM—translocase of the outer membrane and TIM—translocase of the inner membrane).

When cells are exposed to stressful conditions (oxidative stress, temperature alterations, hypoxia, nutrient unavailability, induction of DNA damage, etc.), alterations occurring in the cytosol affect mitochondria, which also respond to stress with changes in metabolism (for review, see [163,164]).

However, dysfunctional mitochondria are also a stress signal that triggers cellular responses, which include mitophagy and unfolded protein response (UPR) pathway activation [165,166]. Both processes participate in the mitochondrial quality control (MQC) system.

Autophagy is a complex process that involves several steps and multiple protein complexes. Briefly, signals such as starvation, hypoxia, oxidative stress, and ER stress, among others, activate the nucleation and isolation of the membrane (usually from ER but also from the recycling endosome, plasma membrane, or Golgi complex) that forms the phagophore. Several protein complexes are recruited to this structure, expanding it into a sphere around a portion of the cytosol. The cargo (organelle or protein aggregates, for example) is recognized and engulfed by this sphere that can be further sealed, forming the autophagosome. This autophagosome, through interactions with microtubules, is transported to the lysosome, and the fusion of the membranes of the lysosome and autophagosome forms the autolysosome, where the content is degraded (for revision [167,168,169,170]). The process of mitochondrial autophagy is named mitophagy.

Mitophagy is an evolutionarily conserved cellular process for removing dysfunctional or superfluous mitochondria through the lysosome. Closely related is the mitochondrial unfolded protein response (UPR^mt^), a cellular stress response mechanism. It involves the initiation of transcriptional activation programs for mitochondrial chaperone proteins and proteases to uphold proteostasis within the mitochondria. Mitophagy plays an important role in MQC, and its impairment perturbs mitochondrial function and causes progressive accumulation of defective organelles, leading to cell and tissue damage. The activity of renewal of the mitochondria is named basal mitophagy. A programmed mitophagy can also occur during development and cell differentiation, for instance, erythrocyte differentiation, cardiomyocyte maturation, and post-fertilization sperm mitophagy, which leads to exclusive mtDNA inheritance. Finally, mitophagy is also triggered in response to stress caused by hypoxia or starvation (for review, see [171]). The process is initiated by PTEN-induced kinase 1 (PINK1), which usually enters healthy mitochondria, but under mitochondria damage or depolarization, it accumulates at the OMM, where it can recruit and phosphorylate ubiquitin Ub E3 protein ligase Parkin, initiating a series of ubiquitination reactions that culminate in the recruitment of autophagy-related proteins [168,170]. Autophagy can be divided into six steps for comprehension: initiation and phagophore nucleation, which involve the ULK (unc-51-like kinase 1) complex and Beclin/VPS (vacuolar protein sorting) complex, phagophore expansion, promoted mainly by ATG (autophagy-related protein) family, cargo sequestration, membrane sealing, autophagosome maturation, and fusion with the lysosome (for review, see [170]) (Figure 7). Among the proteins involved in the autophagic process, the main players are the ubiquitin-like modifiers, which initiate the autophagosome formation and are essential for the later formation of the autophagosome. Briefly, Atg12 is first modified by Atg7, then transferred to Atg10 and conjugated to Atg5. Further, the microtubule-associated protein 1 light chain 3 (LC3I) localized in the cytosol is activated by Atg7, transferred to Atg3, and conjugated to phospholipid (LC3II). LC3 II is considered a marker of autophagosome formation; however, there are other ubiquitin-like modifiers such as GATE-16 (Golgi-associated ATPase enhancer of 16 kDa) and GABARAP (GABA type A receptor-associated protein), which can undergo the same process as LC3, although their role is still less known (for review, see [172]).

For mitophagy, the damaged regions in the mitochondria are first recognized and then fissioned, and they occur mainly to protect the cell from death since the damaged mitochondria are eliminated. In addition to mitochondrial fission induction, fusion-related proteins such as Mfn2 and OPA1 are ubiquitinated by Parkin and degraded by proteasome.

The other MQC mechanism is related to mitochondrial proteostasis. When mitochondria are dysfunctional, the import of proteins is decreased and the accumulation of these proteins at cytosol leads to unfolded protein response in the ER (UPR^ER^). The UPR is a way to communicate with the nucleus to increase the expression of chaperones. A mitochondrial-specific UPR (UPR^mt^) also occurs [173,174].

Inside the mitochondria, chaperones such as heat shock protein family A member 9 (HSPA9 or HSP70), heat shock protein 60 (HSP60), heat shock protein 10 (HSP10), and TRAP1 (heat shock protein 75 kDa or HSP90 mitochondrial) act to help proteins fold correctly or restore the conformation when a misfolded protein is present. When the levels of misfolded proteins increase, the expression of these chaperones is upregulated. Simultaneously, the expression of proteases such as caseinolytic protease proteolytic (CLPP), yme1-like 1 ATPase (YME1 L1), and mitochondrial Lon protease-like 1 (LonP1) is also increased. The UPR^mt^ is activated by mitochondrial ROS excess, mtDNA damage, and most of the stressors that interfere with mitochondria proteostasis or function, like ETC (for review, see [175]) (Figure 7).

The accumulation of mitochondrial proteins in cytosol leads to the phosphorylation and activation of EIF2-alpha (eukaryotic translation initiation factor 2A), which, in turn, enhances the translation of three transcription factors: CHOP (C/EBP homologous protein), ATF4 (activation transcription factor 4), and ATF5 (activation transcription factor 5). The activation of these transcription factors culminates in the expression of UPR-related genes [176,177]. ATF4 and CHOP activation has also been observed in UPR^ER^, while ATF5 is specific to UPR^mt^. ATF5 shows mitochondrial and nuclear localization signals, and the increase in the misfolded proteins inside mitochondria inhibits ATF5 entry in mitochondria, leading to an increase in its translocation to the nucleus, where it activates LONP1 expression. Moreover, the heat shock transcription factor 1 (HSF1) binds to mitochondrial single-stranded DNA binding protein 1 (SSBP1) to form a complex that binds to the promoters of the HSP60 gene, HSP10 gene, and mtHSP70 gene, thereby upregulating the expression of chaperone genes (for review, see [175]).

Until now, only Nek9 and Nek2 were directly implicated in autophagy. Yamamoto and colleagues (2012) identified an LC3-interacting region (LIR) in Nek9 and found that Nek9 mediates the recognition of myosin IIA by GABARAP through this region. GABARAP is a protein that recognizes cargo and conjugates it to autophagic membranes. The interaction of GABARAP and Nek9 is important for cilia formation because this regulatory circuit results in the autophagic degradation of myosin IIA, a negative regulator of ciliogenesis [178]. Growing evidence points to the important role of autophagy in regulating ciliogenesis [179,180]. Moreover, Nek9 can phosphorylate LC3B at threonine 50, suppressing p62 autophagy [181].

Nek2 stabilizes Beclin-1, an important player in initiation and phagophore nucleation, through USP7-mediated deubiquitination, thereby enhancing autophagy [182]. Beclin-1 promotes the formation of PI3KC3–C1 and regulates the lipid kinase VPS34 (for review, see [170]). The function of Nek2 in autophagy seems to be important for the development of resistance in multiple myeloma [182].

Interestingly, several autophagy-related proteins were found in Nek1 and Nek5 interactomes (Figure 1).

Martins and collaborators (2021), using a mitophagy detection kit, that can detect the fusion of mitochondria to lysosomal membrane, observed that HAP1 Nek1 KO cells, showed less signal in basal conditions [46]. This suggests that there may be a deficiency in autophagy activation and considering that Nek1 interactome data show a high number of autophagy-related proteins (4.3%) (Table 1), such as MAP1LC3A/B (also named LC3), GABARAP, ATG5, Beclin1 (BECN1), Parkin (PRKN) ULK2, and SQSTM1 (also called p62), one possible explanation would be that Nek1 is necessary for autophagosome formation (Figure 7). However, whether Nek1 plays a direct role in controlling autophagy must be further investigated.

In 2011, Szyniarowski and colleagues showed that Nek4 depletion in MCF7 cells leads to an increase in LC3 GFP puncta, suggesting a connection between Nek4 and autophagy. However, the results were not conclusive, and the role of Nek4 in the autophagic process is not clear so far [183]. In the Nek4 interactome, LRRK2 and ATG2 were found. Leucine-rich repeat kinase 2 (LRRK2) is an important protein in Parkinson’s disease development, and multiple pieces of evidence point out the role of this protein in several steps of autophagy. Although the mechanism is not completely understood, LRRK2 mutation has been associated with the regulation of autophagosome formation, autophagosome and lysosome fusion, lysosomal maturation, maintenance of lysosomal pH and calcium levels, and lysosomal protein degradation (for review, see [184]). ATG2 was found to be essential for autophagosome formation and lipid droplet size and number [185]. Although we do not have enough information regarding the involvement of Nek4 in autophagy, our results, which showed the overall mitochondrial health and increase in fission promoted by Nek4 overexpression, suggest that Nek4 can play a role in mitochondrial quality control (Figure 7) [20]

In the Nek5 interactome, several proteins related to autophagy were found (ATP6V1E1, SH3GLB1, UBQLN1, UBQLN2, UCHL1, RAB1A, CHMP4B, HMGB1, and VPS4B). Among them, ubiquilin (UBQLN) proteins 1 and 2 were found [18]. The role of ubiquilins in autophagy was presented in the autophagosome acidification, and the depletion of UBQLN delays the delivery of autophagosomes to lysosomes [186]. Moreover, it was demonstrated that UBQLN proteins regulate the vacuolar ATPase (V-ATPase) function [187,188] (Figure 7). Also, the AAA-ATPase vacuolar protein sorting-associated4 B (VPS4B), a component of endosomal sorting complex required for transport (ESCRT), was found to be essential for autophagosome membrane closure [189].

Regarding stress responses, TNF receptor-associated protein 1 (TRAP1, also known as HSP75 or mtHSP90) was found in the interactomes of Nek4 and Nek5 (Figure 1). TRAP1 has been found in the mitochondrial matrix and has already been related to the cellular response to ROS levels [190] and UPR. Amoroso and colleagues (2012) showed that TRAP1 interacts with TBP7/Rpt3, a S6 ATPase and a component of the proteasome in the ER-mitochondria contacts, where both control the protein quality before entry into the mitochondria [191]. The reduction in TRAP1 expression was associated with changes in mitochondrial morphology and a reduction in the expression of fission proteins (Drp1 and MFF) [192].

The prefoldin subunit 2 (PFDN2), identified in the Nek10 interactome, Ref. [19] is a chaperone first described by Vainberg et al. (1998) [193]. The complex formed by PFDN2 and four other subunits, named prefoldin, prevents the misfolding of newly synthesized peptides and protein aggregation. The disruption of prefoldin formation is associated with alpha-synuclein and huntingtin aggregates and cell death [194,195].

The Nek6 interactome retrieves the matrix metalloproteinase 2 (MMP2) and ATF4, both related with cellular response to stress [23]. MMP2 is a mitochondrial protein known by cleaved mitochondrial proteins in response to stress [196].

The interaction with ATF4 was confirmed in vitro, but the consequence of this interaction has not yet been studied. As ATF4 does not show a consensus phosphorylation motif for Nek6, one could suppose that both proteins act in complex in response to stress [23].

## 8. Mitochondrial versus Cell Cycle-Related Functions

Cell division is a complex and highly coordinated event that ultimately leads to DNA duplication and its segregation into two daughter cells mediated by the microtubule organizing activity of the centrosome (Figure 8). Cell division also requires coordination with the energy production machinery to maintain energy supply according to the cell demand during cellular division, which also demands effective communication between the nucleus and the mitochondria. The mitochondria-nuclear communication ensures sufficient energy supplies through the dynamic remodeling and biogenesis of the mitochondria network and regulation of the metabolic rate. Also, the microtubule organizing centers (MTOCs) further mediate the distribution of cell organelles such as mitochondria during cell division (reviewed in [197]). However, the exact mechanisms of how the mitotic machinery controls energy production remains unknown. Some mitotic kinases have already been described to regulate energy production by the direct regulation of mitochondrial proteins.

Mitochondria are not formed de novo and must be distributed to both daughter cells evenly. Studies using high-resolution images in synchronized cells showed that mitochondrial morphology changes considerably during the cell cycle [198] (Figure 8).

A mix of round/oval, small, and big filamentous populations of mitochondria is observed in interphase (Figure 8, step 7). During the G1-S transition, the mitochondria appear in a giant tubular network (Figure 8, step 1 and 2). This morphology is accompanied by a hyperpolarized state and an increase in oxygen consumption and ATP production [198], while, in the early mitotic phase, predominantly small/round mitochondria are observed (Figure 8, step 3–6). Filamentous mitochondria are observed in daughter cells [72] (Figure 8, step 7).

Mittra and colleagues (2009) demonstrate that the hyperfusioned mitochondria state is associated with high levels of cyclin E [198], although the mechanism of this regulation is not known so far. During early mitosis, the pro-fission protein DRP1 is phosphorylated by CDK1/cyclinB, resulting in a distribution by the cytosol of small round mitochondria [69] (Figure 7, step 3).

CDK1, in addition to being the master regulator of mitochondrial fission during mitosis, was also demonstrated to relocate to mitochondria and boost ATP production through the phosphorylation of complex I in order to fuel the cell cycle transition G2/M [199]. Moreover, it was demonstrated that after ionizing-radiation induced DNA damage, CDK1 translocates to the mitochondria and phosphorylates complex I to boost ATP generation, which, according to the authors, would be necessary for providing energy for DNA repair [200]. CDK2 is directly regulated by mitochondrial ROS-mediated oxidation to drive progression through the S phase [201]. Also, CDK4/CDK6 pharmacological inhibition increases mitochondrial metabolism through the elevated utilization of glutamine and fatty acid oxidation in melanoma cells [202].

Aurora kinase A (AURKA) is also required for correct mitochondria fission pre-mitosis. AURKA phosphorylates RalA, in turn, favors DRP1 phosphorylation by CDK1, mediating mitochondrial fission [203]. In interphase, AURKA is also localized in the mitochondria, where it controls mitochondrial fusion by inhibiting DRP1 binding to MFF, consequently regulating ATP production [204].

In addition to the role of regulating mitotic mitochondrial fission, some cell cycle-related kinases also show a role in mitochondrial distribution through the regulation of motor proteins. Mitochondrial morphological changes and movements are regulated by microtubules, actin filaments, and myosin [205]. A recent study described that mitochondrial movement during mitosis is regulated for three different pools of actin: actin cable network, actin waves, and actin comet tails. The action of actin waves and comet tails is important to disperse different types of mitochondria and mix all the content for future equality distribution. Because of this movement, damaged mitochondria are distributed equally among daughter cells [206].

CDK1 and AURKA work in coordination to phosphorylate mitochondrial and cytoplasmic proteins that in turn shed kinesin and dynein from the mitochondrial surface, thus promoting the mitochondrial detachment from microtubules and the distribution in the periphery of the mitotic spindle [198]. Furthermore, centromere protein F (CENP-F) interacts with Miro (reviewed in [73,207]), a Rho-GTPase important for mitochondrial transport and distribution via microtubule and actin [208]. Miro recruits CENP-f to mitochondria during cytokinesis to mediate the association of mitochondria with microtubules [207].

In addition to providing energy, mitochondria can influence the cell cycle by centrosome homeostasis (reviewed in [191,209]).

Centrosome amplification leads to mitochondrion displacement, which is associated with high levels of tubulin acetylation and kinesin 1-mediated transport. Moreover, mtDNA depletion is associated with centrosome duplication defects and increased levels of PLK4 and AURKA [210].

An important player in cell cycle regulation is the primary cilium that is assembled in G0/G1 from a centriole and plasmatic membrane among other components and disassembled in G2/M. The disassembly of the primary cilium is necessary for cell division [211,212] (Figure 8, step 7). The ciliary growth is affected by mitochondrial function in several types of cells, i.e., fission and a decrease in mitochondrial respiration stimulate cilium formation, while fusion is associated with the suppression of this process [213]. Ciliogenesis has been observed when there is an increase in mtROS, AMPK activation, and autophagy induction. When ciliogenesis is inhibited, autophagy is not observed, but there is an increase in cell death [213]. The reduction in ATP levels and complex I activity lead to an increase in cilium length [214,215]. The first results relating to cilia and autophagy were published in 2013, where Tang and colleagues (2013) demonstrated that autophagy triggers the biogenesis of the primary cilium [216], and on the other hand, Pampliega and colleagues (2013) showed that the genetic inhibition of autophagy enhances cilia-associated signaling such as Hedgehog (Hh) signaling under serum starvation [217].

One of the main players in autophagy regulation, PINK1, is essential for mitotic progression, and its depletion leads to an impairment in cytokinesis, causing multinucleated cells and mitochondrial fragmentation. This phenotype is likely to be related to an increase in DRP1 phosphorylation by CDK1 [218].

Although Neks are not required for mitotic entry, Nek2, Nek5, Nek6, Nek7, and Nek9 regulate important processes that take place from interphase to mitosis, including cell division events, centrosome duplication, separation, and disjunction [9,10,219]. Nek2 [220,221,222] and Nek5 [223] control centrosome disjunction (Figure 8, step 5), which facilitates mitotic spindle pole separation. Additionally, Nek9 phosphorylates Nek6/Nek7, which activates kinesin EG5 and consequently, regulates mitotic spindle formation (Figure 8, step 6) [6,224]. Mitotic kinases, cyclin-dependent kinase 1 (CDK1) and serine/threonine-protein kinase (PLK1), phosphorylate Nek9 at the onset of mitosis [225]. Nek2 also regulates kinetochore integrity during mitosis (Figure 8, step 6) [226].

Nek1, Nek4, and Nek10 do not show a clear role in cell cycle regulation. White and Quarmby (2008) showed that Nek1 is important to maintain centrosome stability [227], which is necessary for cell cycle progression, and the reduction in Nek1 expression results in S-phase arrest [49]. The direct effect of Nek1 in cell cycle control is still not clear because of some conflicting observations [49,154], but it has been robustly demonstrated that upon DNA damage, Nek1 is necessary for the G2/M checkpoint [49,154] (Figure 8, step 4).

The studies regarding Nek4, so far, show no effect on cell cycle control [158,228,229]. Although always performed in unsynchronized cells, changes in cell cycle were only observed after DNA damage events [125]. However, considering, for instance, the importance of cilia for cell cycle progression and Nek4’s role in primary cilia stabilization [228], Nek4 may indirectly control the cell cycle through cilium maintenance. Nek10 also only interferes with cell cycle progression after DNA damage [8,37]. The DNA damage response is also important for cell cycle progression once it is part of checkpoints and can drive cell-to-cycle progression or arrest and repair or cell death.

All processes can be connected to mitochondrial function, mainly because all of them require energy but also because of their cell death and autophagy functions.

The communication between mitochondria and the nucleus is very important for controlling cellular response to stress or increasing ATP production for repair, survival signaling, or cell death. P53 is an important player in this communication, regulating the transcription of cell cycle regulators, apoptosis, repair, or survival genes. P53 is located in both mitochondria and the nucleus. P53 also directly regulates the transcription of the fusion protein, Mfn2 [230]. Furthermore, mitochondrial p53 can regulate mtDNA transcription through the interaction with Pol gamma in response to mtDNA damage, and the loss of p53 is associated with an increase in mtDNA damage susceptibility [231]. The role of p53 in autophagy has also already been demonstrated, although is not completely understood. Nuclear p53 increases autophagy by the transcription of autophagy-related genes. On the other hand, cytoplasmic p53 can bind to Parkin, avoiding its translocation to damaged mitochondria and impairing autophagy [232,233]. The role of p53 in autophagy is likely to be related to glycolysis status in the cell [234].

## 9. Conclusions

The main function that has defined the Nek family is the control of the cell cycle, including mitosis and centrosome disjunction. However, we and others have found different roles for the members of this family that are not directly linked to cell cycle control. Although cellular events are connected and occur simultaneously in the cell, the mitochondrion is an organelle that reproduces independently and even replicates its own DNA. However, its function influences the whole cell and in turn, also depends on the nuclear cycle and several proteins encoded by nuclear genes. We are working to identify the biological functions of members of the Nek family that do not present classic participation in cell cycle regulation, and we have found that several members are important players in mitochondrial function. The first evidence for a mitochondrial function appeared for Nek1 and at the same time, in our interactome studies, we have observed several mitochondrial partners for Nek4 and Nek5. Our findings have demonstrated that mitochondrial regulation is a new axis in Nek family functions and that several members can cooperate on different mitochondrial processes. While Neks 1, 4, 6, and 10 seem to stimulate mitochondrial respiration, Nek5 decreases mitochondrial respiration. The knockdown of Nek1, 6, and 10 leads to mitochondrial fragmentation, and on the other hand, Nek4 knockdown causes mitochondria hyperfusion. Moreover, Nek1, Nek4, Nek5, and Nek10 play a role in mtDNA maintenance. It is interesting to note that, considering the interaction partners, each Nek seems to be more implicated in a specific process: Nek1 in autophagy, Nek5 in mtDNA maintenance and stress response, Nek4 in mitochondria dynamic, and Nek10 in metabolism.

## Figures and Tables

**Figure 1 cells-13-00473-f001:**
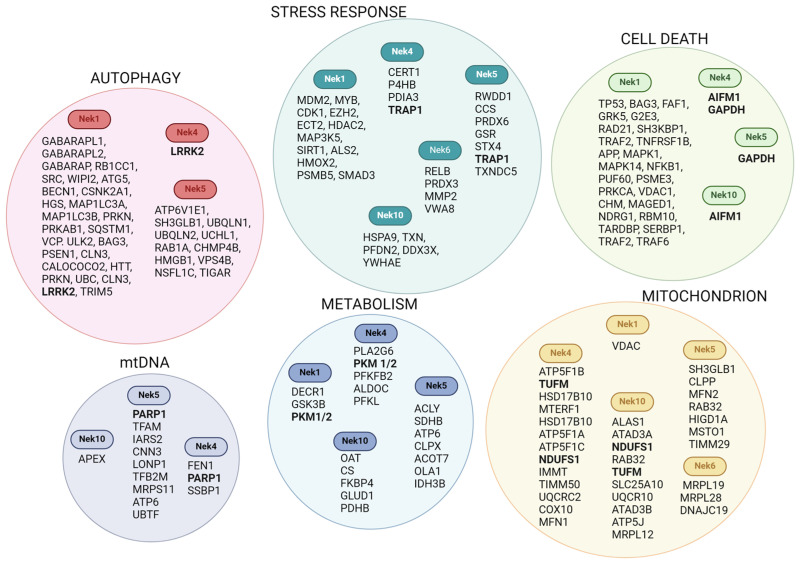
Proteins from Nek interactomes, according to mitochondrion-related functions. Proteins from Nek1 (untreated samples [22]), Nek4.1 (isoform 1 [13]), Nek5 [16,18], Nek10 [19], and Nek6 [23] interactome were analyzed using the Functional Annotation Chart from DAVID [24]. Proteins found in the processes related to mitochondrial functions (apoptosis/ cell death, autophagy, stress response, mitochondrial dynamics or structural function, and mtDNA integrity maintenance or metabolism) were separated and are presented with their gene abbreviations in the respective group. Proteins found in the interactome of more than one Nek are shown in bold. Some proteins were manually added based on information present in the literature. Cell death: positive regulation of apoptosis and negative regulation of apoptosis. Stress response: response to reactive oxygen species (ROS) and unfolded protein response (UPR). Mitochondrion: mitochondrial membrane components and mitochondrial fission/fusion control. mtDNA: DNA repair and transcription regulator. Metabolism: proteins from glycolysis and lipids metabolism. Created with BioRender.com (accessed on 23 January 2024).

**Figure 2 cells-13-00473-f002:**
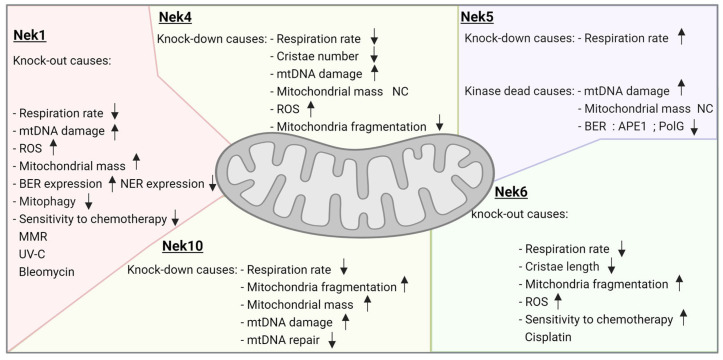
Neks 1, 4, 5, 6, and 10 all are involved in the regulation of mitochondrial homeostasis. Knockout (KO), knockdown Neks 1, 4, 6, and 10, or the expression of a Kinase dead variant (Nek5 only) all cause changes in mitochondrial respiration, mitochondrial dynamics, mtDNA damage or repair, apoptosis, ROS levels, and morphologic phenotypes. While the depletion of Nek1, Nek4, Nek6, and Nek10 decreases mitochondrial respiration, Nek5 knockdown increases respiration rate. Mitochondrial fragmentation was observed in Nek1, Nek6, and Nek10 KO or knockdown cells, and Nek4 knockdown promotes hyperfusion. This figure sums up the major effects of the knockout/down. The up-arrow heads indicate upregulation or increase, whereas the down-arrow heads indicate downregulation or decrease in the functional aspect in question. Created with BioRender.com (accessed on 23 January 2024).

**Figure 3 cells-13-00473-f003:**
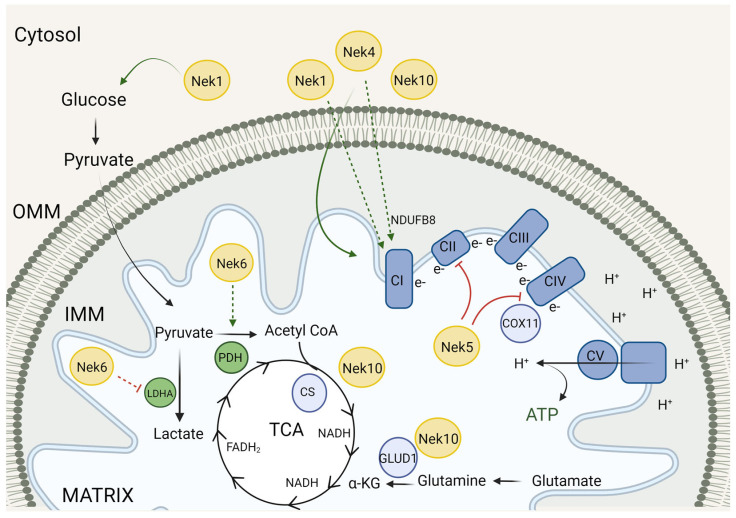
Neks regulate mitochondrial metabolism. Neks 1, 4, and 10 increase mitochondrial respiration, and Nek5 decreases respiration. Nek5 overexpression is associated with CII and CIV complexes activity reduction, and Nek5 interacts with matrix protein COX11. Nek10 is related to the tricarboxylic cycle and can interact with GLUD1, which is important for glutamate and glutamine metabolism for generating alpha ketoglutarate that enters TCA. Also, Nek10 can interact with CS regulating its activity. Nek4 at MOM regulates mitochondrial morphology that induces mitochondrial fission and increases mitochondrial respiration, accompanied by an increase in CI subunit expression. Consequently, Nek4 increases mitochondrial respiration and ATP production. Nek1 also increases mitochondrial respiration and complex subunit expression (complex I). Nek1 also seems to contribute positively to glucose uptake and metabolism. An increase in the LDHA and a decrease in the PDH levels was observed in Nek6 knockout cells. It is expected that Nek6 increases acetyl-coA and inhibits lactate production. Green arrows indicate activation and red lines indicate inhibition related to a Nek putative effect. Dashed lines indicate changes in protein expression. Light blue circles represent putative or confirmed interactors of the Neks. Created with BioRender.com (accessed on 23 January 2024).

**Figure 4 cells-13-00473-f004:**
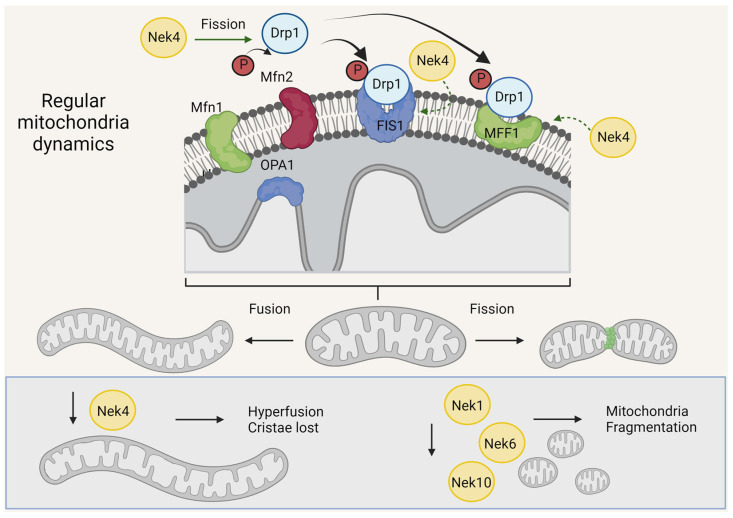
Nek1, Nek6, and Nek10 show an opposite effect in mitochondrial morphology compared to Nek4. Mitochondrial morphology is shaped by the processes of fusion and fission. In unsynchronized cells, there is a mixture of different mitochondrial morphologies. Tubular elongated and small round mitochondria can be observed. The main players controlling mitochondrial morphology are fusion proteins (Mfn1, Mfn2, and OPA1) and fission-related proteins, namely DRP1 and its adaptors MFF, Fis1, and Myd49/50 (not presented). While Nek4 loss of function promotes mitochondrial dysfunction related to the hyperfusion phenotype, Nek10 depletion is associated with an impairment in mitochondrial function and highly fragmented mitochondria. The Nek4 effect is directly related to mitochondrial dynamic proteins, while the fragmentation induced by Nek10 depletion seems to be a secondary effect of a mitochondrial metabolism defect. Nek4 promotes an increase in DRP1 phosphorylation and fission. Nek6 knockout prostate cancer cell line Du145 also shows fragmented mitochondria, but the mechanism is still under investigation [21]. The Nek1^Kat2J/Kat2J^ mouse embryonic fibroblasts also show fragmented mitochondria. The Nek4 effect is directly related to mitochondrial dynamic proteins, while the fragmentation induced by Nek10 depletion seems to be a secondary effect of a mitochondrial metabolism defect. Nek4 promotes an increase in DRP1 phosphorylation and fission. The mechanism for morphologic changes induced by Nek1 and Nek6 depletion is not known so far. Dashed lines indicate changes in protein expression. Light blue circles represent putative or confirmed interactors of the Neks. Black arrow pointing down indicates reduced protein expression. Created with BioRender.com (accessed on 23 January 2024).

**Figure 6 cells-13-00473-f006:**
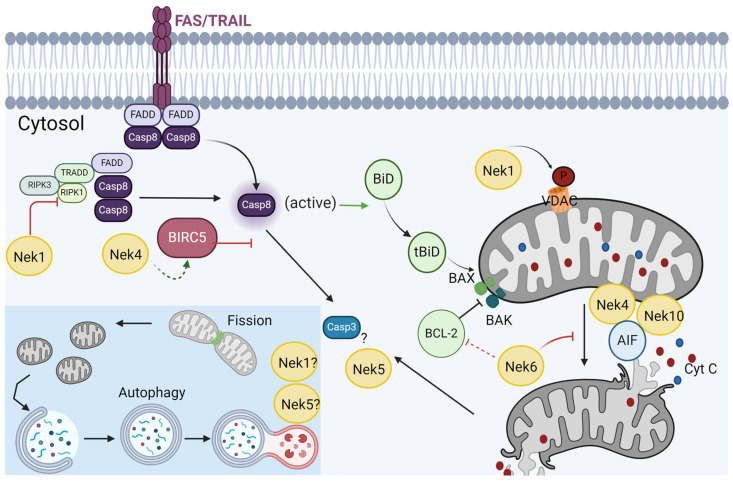
Neks affect death/survival pathways in the cell. Depending on cellular status, cells can activate death or survival pathways. The classical intrinsic apoptosis activation involves cytochrome c release, Apaf-1, and caspase activation. The release of mitochondrial matrix content is caused by the loss of mitochondrial membrane integrity with the opening of pores by Bax and Bak. Nek1 has already been implicated directly in intrinsic apoptosis by VDAC phosphorylation, keeping it in a close state, maintaining mitochondrial membrane potential, and preventing apoptosis. Nek1 also participates in extrinsic apoptosis or necroptosis pathways by inhibition of RIPK1 pool formation. RIPK1 can be activated by TNFa/TNFR signaling. Also, in some cell types, RIPK1 and RIPK3 can be activated by mtROS, which triggers RIPK1 autophosphorylation and necrosome formation [148,155]. Also, Nek1 could participate in autophagy due to a vast number of interactors related to this process; however, this must be further investigated. Related to the extrinsic apoptosis pathway, the activation of death receptors by ligands such as TNFa, Fas, TNFRS10, and TRAIL triggers the activation of caspase-8, and then -3, and -7. Nek4 has already been implicated in this pathway because it induces BIRC5 expression and inhibits TRAIL-induced cell death. Nek5 is a substrate of caspase 3 in mice muscle cells, and in these cells, the overexpression of Nek5 causes an increase in apoptosis. On the other hand, in oxidative conditions, HEK293T Nek5-overexpressing cells are resistant to death. AIF1 was found to be an interactor in both the Nek4 and Nek10 protein interactome analysis. If this interaction were confirmed, this would suggest that these proteins could participate in the communication between the mitochondria and the nucleus, especially in cases of cellular stress. Nek6 KO seems to decrease BCL-2 antiapoptotic levels and sensitize cells to apoptosis. Arrows indicate activation, interaction, or functional regulatory interference. Dashed arrows indicate changes in protein or mRNA expression. Red lines indicate inhibitory regulatory influence, and green lines indicate positive influence. Light blue circles represent putative interactors of the Neks. Created with BioRender.com (accessed on 23 January 2024).

**Figure 7 cells-13-00473-f007:**
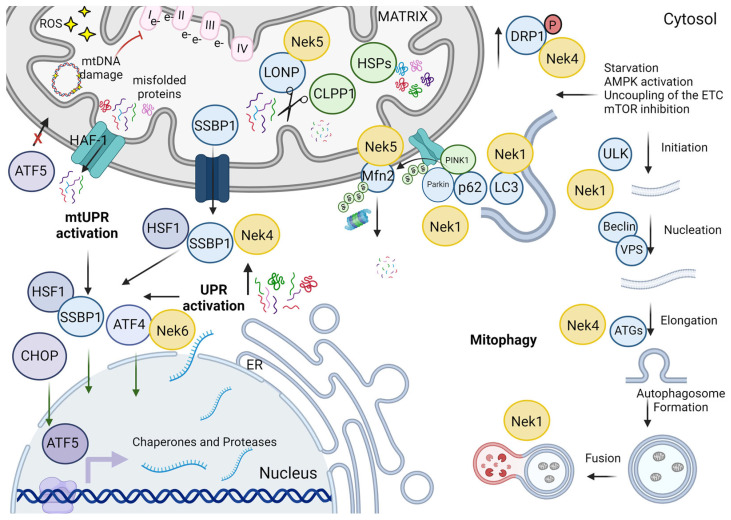
Neks participate in the regulation of mitophagy and the unfolded protein response, both being pathways of mitochondria quality control. Simplified unscaled representations of mitophagy and UPR (UPR^ER^ and UPR^mt^) are shown on the right and the left of the figure, respectively. On the right side, the stress stimuli, such as starvation, mitochondrial respiration uncoupling, AMPK pathway activation, or mTOR inhibition lead to the initiation of lipid recruitment by the ULK complex. Nucleation is mainly mediated by the Beclin/VPS/ATG complex. The elongation of autophagosomes is mediated by the ATG family. Simultaneously, in the MOM, proteins like BNIP3, NIX, FUNCD (not presented in the figure), and PINK1 are stabilized in the mitochondrial outer membrane. PINK1 recruits Parkin, which promotes the ubiquitination of several mitochondrial proteins that will be then phosphorylated by PINK1. These signals are recognized by adaptor proteins, such as OPTN, NPPS2 (not presented), and p62 (SQSTM1). These proteins initiate LC3 recruitment and consequently, the autophagosome membrane. Also, the mitochondrial fission is activated, and the fusion is inhibited by Mfn2 and OPA1 degradation. On the left side, the UPR^ER^, an ER response to the accumulation of misfolded proteins, and the UPR^mt^ are shown. The UPR^mt^ can be activated by an increase in the levels of mtDNA damage, ROS levels or dysfunctional respiration. All these processes are related to disturbed mitochondrial proteostasis. The responses to UPR^mt^ and UPR are related to the activation of ATF4, ATF5, and CHOP transcription factors. The regulation of ATF5 is not completely understood, but it is the transcription factor exclusive to UPR^mt^. In stress conditions, ATF5 importation to mitochondria is inhibited, favoring its nuclear activity. Also, the SSBP1, a mitochondrial matrix protein, exits from mitochondria, and at the cytosol, it interacts with HSF1, and the complex translocates to the nucleus, where it activates the transcription of chaperones. Inside the mitochondria, the proteases, LONP and CLPP1, cleave misfolded proteins that can be exported to the cytosol, activating the ER UPR pathway. The final effect of UPR^mt^ and UPR^ER^ activation is related to the increase in the expression of chaperones, represented by HSPs (such as HSP760, HSP60, HSP10, and TRAP1), and proteases, such as LONP1 and CLPP-1. Possible interactors of Neks are presented in blue (Beclin, VPS, LC3, etc.). The transcription factors are shown in violet, except ATF4, which is an interactor of Nek6. Created with BioRender.com (accessed on 23 January 2024).

**Figure 8 cells-13-00473-f008:**
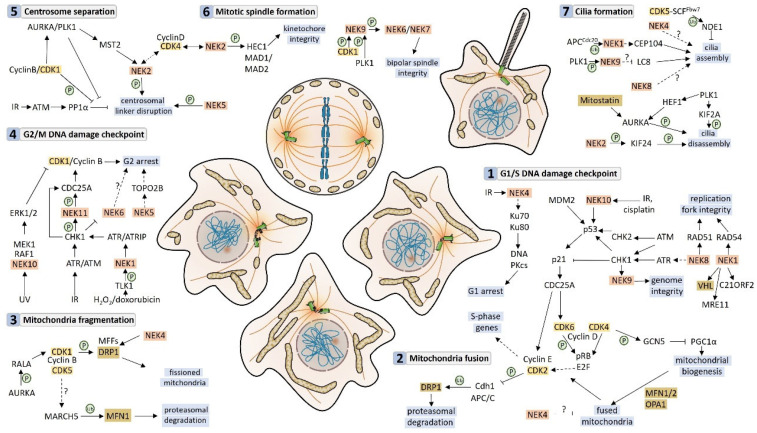
Crosstalk between mitochondrial and other functions of Neks throughout the cell cycle. The different “Nek-related” events occurring during the cell cycle are divided into 7 “steps” and are organized in a clockwise fashion (starting on the right side of the figure): step 1 = G1/S DNA damage checkpoint, step 2 = mitochondrial fusion, step 3 = mitochondrial fragmentation, step 4 = G2/M DNA damage checkpoint, step 5 = centrosome separation, step 6 = mitotic spindle formation, and step 7 = cilia formation. Created using Inskscape software (https://inkscape.org/ accessed in 3 September 2022).

## Data Availability

Data sets mentioned in this paper will be provided upon reasonable request.

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
