# Peer review of "The Mitochondrial Connection: The Nek Kinases’ New Functional Axis in Mitochondrial Homeostasis"

_cells, 2024, doi:10.3390/cells13060473_

Round 1

Reviewer 1 Report

Comments and Suggestions for Authors

Comments on the Quality of English Language

included in first attachment

Author Response

Reviewer #1:

“This review paper highlights work that links members of the NEK kinase family to various elements of mitochondrial biology. I commend the authors for the work done and this attempt

to organize vast amounts of information. This paper is full of useful content, broken down into sections that highlight different elements of mitochondrial biology such as mitochondrial metabolism, mitochondrial morphology, mitochondrial DNA maintenance, mitochondrial response to stress and cell death, response to stress and autophagy, and the relationship between mitochondria functions and the cell cycle. The authors provide some useful figures that seek to highlight NEK roles, measured and implied, in these different areas. The review is a great source of references for the interested reader. This is an exciting area of study and much needs to be unraveled still. For these reasons, the paper will be a useful resource for the community and a useful addition to the literature.”

Response: We thank the reviewer for the positive comments on the manuscript. We agree that this revision is a great source of useful information for the researchers in the field and also the area of research is exciting.

“There is substantial work that needs to be done, however, to get this ready to be published. To this reader, much of the language was confusing, so I think a good deal of proof reading and rewriting needs to take place. Here I tabulate some of the areas that were confusing to me or just need correcting and language alteration. I likely missed some. A careful rewrite needs to be undertaken to make the paper more readable and thus accessible. This is worth doing because

the authors have highlighted very interesting NEK/mitochondria links that deserve further study.”

We agree that the review needs a grammar revision and some changes to make the paper more readable. We thank the reviewer for the extensive work in pointing out our mistakes and confusing sentences. He/she points out that the style and language need some major corrections.

The reviewer points to a 3 pages long list of smaller grammar and orthographic corrections. Please note we carefully revised the whole manuscript and corrected the mistakes to improve the English. Thanks for all these nice suggestions. We think our revision is a lot improved.

Please, see below all the corrections/ suggestions were addressed.

In abstract: What does “besides the primordial one” mean?

We have rephrased the sentence

Introduction: wrong word here “...comprehending coiled coils..” I don’t know what this means.

We have rephrased the sentence.

P2 line 47 “Mitochondrial functions are fine regulated by…” this is awkward

Please note, we have rephrased the sentence.

P3 line 77: briefly provide more detail on VDAC

We have added some information.

P3 line 81: instead of implied do you mean implicated?

We have corrected that.

P3 line 97: Pleas clarify “The last ones usually reach mitochondrial…” I don’t know what the last

ones means.

We have rephrased the sentence

P3 line 109: “shows a considerable score for mitochondrial exportation” needs to be rephrased

for clarity. Also, please mention the scoring methods here, not just in the figure legend,

otherwise these numbers don’t have much meaning. Explain (briefly) these calculations.

Please note we have rephrased the sentence, and we are now providing more information about the predictors. Current page 4 lines 120-129??

P3 line 124: “centeer” is a misspelling.

We have corrected the word.

P4 line 131: I think you mean “in depth” rather than “in deep”

We have corrected the word.

P4 table 1: this table needs more explanation. Describe exactly what is meant by “% of genes)

Please note we have added more details in the caption

P4 line 139: mention that BP is biological process abbreviation

We have added the information

P4 line 142/143: this sentence “NEK6 gene list did not show none…” to get rid of the double

negative and make it clear for the reader

We have corrected that.

P5 line 151: rewrite MitoFates sentence because it is not clear to me what you mean.

We have rephrased the sentence

P5 1st paragraph of section 2: Needs rewriting as it is phrased awkwardly and difficult to read for

the meaning you intend.

We have rephrased the sentence

P5 line 167: “Five complexes participate of OXPHOS…” this is unclear.

We have corrected that.

P5 line 174 Please rewrite this sentence: “The metabolism of glutamine…integrates the TCA” I

do not understand it as written.

We have rephrased the sentence

P5 line 198/199: The last sentence does not make sense to me and needs rewriting for clarity.

We have rephrased the sentence

P7 line 258: This paragraph needs rewriting to make it clearer. For example “Unlike, there is an

increase of proton leak…” is confusing.

We have rephrased the sentence

P9 paragraph starting at line 357: needs re-writing for clarity

We have rephrased the sentence

P10 Figure 3 legend: explain arrows (dashed or not dashed?)

We have added the information

P10 line 368: should be “can interact”

We have corrected that.

P10 section 3: I think a better explanation/definition of what you mean by mitochondrial

morphology would be appropriate at the beginning of this section. You get at it a bit in the

figure, but I think an introduction would help.

Please note we have added some information. Current page 13.

P11 line 447: sentence that begins with “It was also observed…” needs rewriting for clarity.

We have rephrased the sentence

P13: legend for Figure 5 is hard to understand. “In one interphasic cell, from a mitochondrial

tubular network…” I don’t understand this sentence.

We have rephrased the sentence

P14 line 553 “transcription” is misspelled.

We have corrected that.

P14 line 558 “is not straight full” I don’t know what this means.

We have corrected the word.

P15 line 608: the sentence/section that starts with “As Ber is the main pathway…” is confusing,

and I suggest rewriting.

We rewrite the sentence

P16 legend for figure 5: the description in the legend is very confusing and I suggest rewriting.

Please also explain dashed vs solid arrows.

We have explained the meaning of the arrows

P19 section 6 line 83: “composted” is the wrong word here

We have corrected that.

P19 line 812: “…and well described responses comprehend autophagy…” I don’t understand this

sentence.

Please note we rewrite this section and we have added new information based on the references recommended. We also have added a new figure for this section (Figure 7)

P19 line 814 “Autophagy is a complex process that comprehends several steps…” needs

rewriting – comprehends is not the right work here for the meaning you are trying to convey

We have corrected that.

P20 line 852: What is (ano)??

We have corrected that.

P20 line 855: “because mediates” needs rewriting

We have corrected that.

P20 line 866: I think col must mean collaborators, but you should just write that out.

We have corrected that.

P22 Figure 6 caption: The part of the caption in bold and the next part “and caspases

activation” that is not in bold don’t make sense as written.

We have removed the bold text

P22 line 924: this sentence about Nek1 needs to be rewritten for clarity.

P22 line 932. I don’t understand the last sentence of the figure 6 caption.

We have rephrased the sentence

P23 line 948. I suggest rewriting the sentence that starts with “Firstly, as mitochondria…” as I

find it difficult to understand.

We have rephrased the sentence

P24 line 1007: please rewrite sentence that begins “Moreover, ciliogenesis seems…” to make

your meaning clearer

We rewrite the sentence

P25, figure 7 legend: “ordinated” is not the correct word here

We have corrected the word.

P25 line 1066 “anusript” should be “manuscrip

We have corrected the word.

Reviewer 2 Report

Comments and Suggestions for Authors

Authors presented the review of Nek kinase family and their new functions with mitochondria.

Since many evidences are pointing to the connections between neurodegenerative diseases with mitochondria, it would be interesting for readers to consider to investigate the Nek kinases.

Since Nek kinases seemed to have many functions with many organelles, authors should include several indicators to investigate the role of Nek kinase in each implications towards their functions or organelles.

Author Response

“Authors presented the review of Nek kinase family and their new functions with mitochondria. Since many evidences are pointing to the connections between neurodegenerative diseases with mitochondria, it would be interesting for readers to consider to investigate the Nek kinases.”

Response: We thank you very much for the insightful comment. Indeed, a very promising field in mitochondria-related disease is the study on neurodegeneration. Many publications have already demonstrated the importance of mitochondria dynamic, autophagy regulation, and unfolded protein response for diseases such as Parkinson, Alzheimer, Huntington and also amyotrophic lateral sclerosis (ALS) [1–4]. We believe that Neks can be potential candidate to regulate mitochondrial function with an impact on neurological functions. Indeed, several reports have found Nek1 mutations in ALS samples but until now, the mechanism presented a relationship with microtubules and nuclear importation. However, considering all the evidences of an important role of Nek1 in mitochondrial function we believe that worth investigating if Nek1 also can be implicated in the development of ALS through mitochondria function [5–8]. Some evidence also indicates that the inhibition of Nek6 could decrease the toxicity of dipeptide repeat proteins found in ALS by restoration of DNA damage response by p53 [9]

Since Nek kinases seemed to have many functions with many organelles, authors should include several indicators to investigate the role of Nek kinase in each implications towards their functions or organelles.

The reviewer suggests we should investigate and indicate the implications of Neks in organelle functions at large (if we understood correctly).

We also believe that Neks show a role in multiple organelles and is worth investigating and comparing the role of different Neks members in other organelles. However, until recently the canonical role of this family of proteins was more related to cell cycle control and the information regarding their molecular function in other organelles is still scarce. Other reviews are addressing the role of Nek1, Nek8, Nek4 on cilia formation [10–13]. Our focus in this review is to show a completely novel role for Nek family in the mitochondrion, which in our point of view is a breakthrough for the field of Neks.

We think that future reviews should address the cross-talk of neks roles in the nucleus, mitochondria, cilia, ER, and golgi etc.

Reviewer 3 Report

Comments and Suggestions for Authors In this manuscript, entitled " The mitochondrial connection: Five Nek kinase family members establish regulation of mitochondrial homeostasis as a new functional axis for Neks " authors focus in the effect of Neks function on mitochondria metabolism, morphology, mtDNA maintenance, and mitochondrial autophagy and stress response.
It is a well written and described manuscript which for the first time, if I'm not mistaken, authors present a comprehensive study of the relationship between Neks enzymes and mitochondria.

I believe that the presented review will be useful for mitochondria community and also in cancer research.However,I found several parts of the manuscript to be inadequate (see below).Αρχή φÏŒρμας

Howevh I found several parts of the manuscript to be inadequate (see below).

Major comments

·     -   Authors are encouraged to revise the content in this section “4.1. Neks involvement mtDNA maintenance” to enhance precision regarding the correlation between NEKs and mtDNA maintenance. For example, the paragraph (Lines 577-581)

“A possible role of Nek1 in nuclear DNA repair has already been demonstrated in response do ionizing radiation. In this context, Nek1 colocalizes with γ-H2AX and NFBD1/MDC1 [117]. Further, studies from Liu and colleagues (2013) showed that Nek1 kinase activity stabilizes the complex between the checkpoint kinase ATR (ATM and Rad3- related) and its partner ATRIP (ATR-interacting protein), important for efficient DNA damage signaling [118].”

is referred to Nuclear DNA maintenance.

·      -  In the section “Response to stress and autophagy” authors should mention the terms a) mitophagy which is an evolutionarily conserved cellular process to remove dysfunctional or superfluous mitochondria through lysosome help b) Mitochondrial Unfolded Protein Response (UPRmt) is a cellular stress response mechanism. It involves the initiation of transcriptional activation programs for mitochondrial chaperone proteins and proteases to uphold proteostasis within the mitochondria.  

It would be also beneficial to include a Figure for this section and add insights from below studies

Palikaras K, Lionaki E, Tavernarakis N. Mechanisms of mitophagy in cellular homeostasis, physiology and pathology. Nat Cell Biol. 2018 Sep;20(9):1013-1022. doi: 10.1038/s41556-018-0176-2. Epub 2018 Aug 28. PMID: 30154567.

Picca A, Faitg J, Auwerx J, Ferrucci L, D'Amico D. Mitophagy in human health, ageing and disease. Nat Metab. 2023 Dec;5(12):2047-2061. doi: 10.1038/s42255-023-00930-8. Epub 2023 Nov 30. PMID: 38036770.

Zhu L, Zhou Q, He L, Chen L. Mitochondrial unfolded protein response: An emerging pathway in human diseases. Free Radic Biol Med. 2021 Feb 1;163:125-134. doi: 10.1016/j.freeradbiomed.2020.12.013. Epub 2020 Dec 21. PMID: 33347985.

·        -Authors are advised to incorporate a conclusion that underscores the significance of the association between mitochondria and NEKs.

Minor comments

·        Authors should correct the in spelling mistake line 497 “The effect o Nek4 on mitochondrial morphology could results from its interaction”  to The effect of….

·        Please add a reference to below statement (Lines 466-469) “An increase in mitochondrial fission is usually associated with a decrease in mitochondrial functions (respiration and quality), however, in several tissues, it has already been demonstrated that fission induction is necessary to handle with excess of nutrient and low ATP demand”

·        Please rephrase the sentence in line 808” When cells suffer some kind of stress”

Comments on the Quality of English Language

I found the manuscript to be overall well written and be well described. 

Author Response

Reviewer #3:

In this manuscript, entitled " The mitochondrial connection: Five Nek kinase family members establish regulation of mitochondrial homeostasis as a new functional axis for Neks " authors focus in the effect of Neks function on mitochondria metabolism, morphology, mtDNA maintenance, and mitochondrial autophagy and stress response.

It is a well written and described manuscript which for the first time, if I'm not mistaken, authors present a comprehensive study of the relationship between Neks enzymes and mitochondria. 

I believe that the presented review will be useful for mitochondria community and also in cancer research. However,I found several parts of the manuscript to be inadequate (see below).

Response: We thank you for reviewing our manuscript and by the correction and suggestions that will greatly improve our revision. The reviewer is correct, this revision shows for the first time the relationship between all Neks implicated in mitochondrial function so far.

Major comments

  • - Authors are encouraged to revise the content in this section “4.1. Neks involvement mtDNA maintenance” to enhance precision regarding the correlation between NEKs and mtDNA maintenance. For example, the paragraph (Lines 577-581)

“A possible role of Nek1 in nuclear DNA repair has already been demonstrated in response do ionizing radiation. In this context, Nek1 colocalizes with γ-H2AX and NFBD1/MDC1 [117]. Further, studies from Liu and colleagues (2013) showed that Nek1 kinase activity stabilizes the complex between the checkpoint kinase ATR (ATM and Rad3- related) and its partner ATRIP (ATR-interacting protein), important for efficient DNA damage signaling [118].”

is referred to Nuclear DNA maintenance.

Indeed, we have included the relationship between Neks and nuclear DNA repair to show that for some Neks the involvement in DNA repair have already been reported. Moreover, some proteins that participate in DNA repair are common between mtDNA and nuclear DNA repair. We could emphasize that cited literature deals largely with nuclear DNA. But to avoid some misinterpretation we carefully revised this section, which now reads:

“Although Nek1 so far was not implied directly in the maintenance of mtDNA a study implicates  Nek1 in maintenance and repair of nuclear DNA , especially  in response do ionizing radiation. In this context, Nek1 co-localized with γ-H2AX and NFBD1/MDC1 [117]. Further, studies from Liu and colleagues (2013) showed that Nek1 kinase activity stabilizes the complex between the checkpoint kinase ATR (ATM and Rad3-related) and its partner ATRIP (ATR-interacting protein), important for efficient DNA damage signaling [118].

Although Nek1 so far was not implicated directly in the maintenance of mtDNA, severalstudies implicate Nek1 in maintenance and repair of nuclear DNA [123,124].In this context, Nek1 co-localized with γ-H2AX and NFBD1/MDC1 [123]. Further, studies from Liu and colleagues (2013) showed that Nek1 kinase activity stabilizes the complex between the checkpoint kinase ATR (ATM and Rad3-related) and its partner ATRIP (ATR-interacting protein), important for efficient DNA damage signaling [124].

  • - In the section “Response to stress and autophagy” authors should mention the terms a) mitophagy which is an evolutionarily conserved cellular process to remove dysfunctional or superfluous mitochondria through lysosome help b) Mitochondrial Unfolded Protein Response (UPRmt) is a cellular stress response mechanism. It involves the initiation of transcriptional activation programs for mitochondrial chaperone proteins and proteases to uphold proteostasis within the mitochondria. 

It would be also beneficial to include a Figure for this section and add insights from below studies

Palikaras K, Lionaki E, Tavernarakis N. Mechanisms of mitophagy in cellular homeostasis, physiology and pathology. Nat Cell Biol. 2018 Sep;20(9):1013-1022. doi: 10.1038/s41556-018-0176-2. Epub 2018 Aug 28. PMID: 30154567.

Picca A, Faitg J, Auwerx J, Ferrucci L, D'Amico D. Mitophagy in human health, ageing and disease. Nat Metab. 2023 Dec;5(12):2047-2061. doi: 10.1038/s42255-023-00930-8. Epub 2023 Nov 30. PMID: 38036770.

Zhu L, Zhou Q, He L, Chen L. Mitochondrial unfolded protein response: An emerging pathway in human diseases. Free Radic Biol Med. 2021 Feb 1;163:125-134. doi: 10.1016/j.freeradbiomed.2020.12.013. Epub 2020 Dec 21. PMID: 33347985.

We thank you for this important point that we miss to explore in this section. We have addressed only the general autophagy process and UPR because the mtUPR in mammalians is so far not completely explored. Please note we have included the mention regarding mtUPR and mitophagy specifically and we have included a new figure about mitophagy and unfolded protein response (Figure 7). The recommended references were also included.

 The mentioned section in the text was modified as follows:

        “However, dysfunctional mitochondria are also a signal to cellular response and well described responses comprehend autophagy and unfolded protein response pathways activation . Mitophagy, is an evolutionarily conserved cellular process to remove dysfunctional or superfluous mitochondria, through lysosome help. Closely related is the Mitochondrial Unfolded Protein Response (UPRmt), a cellular stress response mechanism. It involves the initiation of transcriptional activation programs for mitochondrial chaperone proteins and proteases to uphold proteostasis within the mitochondria.  “

Please note we have added more information regarding both processes: mitophagy and mtUPR

  • -Authors are advised to incorporate a conclusion that underscores the significance of the association between mitochondria and NEKs.

Please note, we have included the conclsuion

“The primordial function that has defined the Nek family was the control of the cell cycle (mitosis and centrosome disjunction, for instance), however, we and others have found different roles for the members of this family not directly linked to cell cycle control. Although cellular events are connected and occur simultaneously in the cell, the mitochondrion is an organelle that works somehow independently, since replicates its own DNA. However, its correct function influences the whole cell and also is affected by the nucleus cycle and depends on several proteins codified by nuclear DNA. We are working to identify the biological functions of members of Nek family that do not present the classical participation in cell cycle regulation and we have found that several members are important players in mitochondrial proper function. The first evidence for a mitochondrial function appeared for Nek1 and at the same time, we were observing in our interactome studies several mitochondrial partners for Nek4 and Nek5. Our findings have demonstrated that mitochondrial regulation is a new axis in Nek family functions and that several members can cooperate or, balance different mitochondrial processes. While Neks 1, 4, 6, and 10 seem to stimulate mitochondrial respiration, Nek5 decreases it. The knockdown of Nek1, 6, and 10 leads to mitochondrial fragmentation, and on the other hand, Nek4 knockdown causes mitochondria hyperfusion. Moreover, Nek1, Nek4, Nek5, and Nek10 show a role in mtDNA maintenance. It is interesting to note that considering the interaction partners, each Nek seems to be more implied in a specific process: Nek1 in autophagy, Nek5 in autophagy, mtDNA maintenance and stress response, Nek4 in mitochondria dynamic, and Nek10 in metabolism”

Minor comments

  • Authors should correct the in spelling mistake line 497 “The effect o Nek4 on mitochondrial morphology could results from its interaction” to The effect of….

We have corrected that sentence.

  • Please add a reference to below statement (Lines 466-469) “An increase in mitochondrial fission is usually associated with a decrease in mitochondrial functions (respiration and quality), however, in several tissues, it has already been demonstrated that fission induction is necessary to handle with excess of nutrient and low ATP demand”

Please note that we have cited the revision from Liesa and Shirihai that explore the mitochondrial dynamics in the regulation of nutrient utilization [14]

  • Please rephrase the sentence in line 808” When cells suffer some kind of stress”

Please note that we have rephrased the sentence that is:

      “When cells are exposed to  stressful conditions (oxidative stress, temperature alterations, hypoxia, nutrient unavailability, induction of DNA damage etc. ), alterations occurring in the cytosol affect mitochondria, that also respond to stress with changes in metabolism (For review, [161,162]). “

Minor comments:

The reviewer suggest 3 corrections of words or spelling. We corrected everything accordingly.

Round 2

Reviewer 1 Report

Comments and Suggestions for Authors

Thank you for your careful responses to my initial review. Here are a few minor items I noticed.

P3 line 112: I suggest “For comparison purposed, we have analyzed …as control proteins for these analyses.”

P5 line 173 I suggest “fatty acids” instead of “fat acids”

P13 line 527 “could result” instead of “could results”

Author Response

Reviewer #1:

We thank the reviewer for the careful correction of our manuscript. We have addressed the minor comments as follows:

Minor comments:

P3 line 112: I suggest “For comparison purposed, we have analyzed …as control proteins for these analyses.”

 We have corrected the sentence.

P5 line 173 I suggest “fatty acids” instead of “fat acids”

We agree, and we have corrected that.

P13 line 527 “could result” instead of “could results”

We have corrected the word.

Reviewer 3 Report

Comments and Suggestions for Authors

I would like to thank the authors for their responses.

I have some minor comments

1)Please rephrase the sentence for more clarity line 898-900 "However, dysfunctional mitochondria are also a signal to cellular response and well 898 described responses comprehend mitophagy and unfolded protein response pathways 899 activation [163,164] Both processes constitute the mitochondrial quality control (MQC). "

2) Please correct the sentence line 902: Autophagy is a complex process that involved several steps and multiple protein complexes. 

Autophagy is a complex process that involves several steps...

3)Please rephrase for more clarity the paragraph line 912-line 924: "Autophagy is a vastly studied response of the cell to mitochondrial damage and is named mitophagy......"

Autophagy is not a response specific only for mitochondria, but also for other organelles. Authors define mitophagy twice in this paragraph

4) Keep the same symbols for UPRET and UPRmt throughout the text

5)Please rephrase the sentence line 1145." When ciliogenesis is inhibited no autophagy is observed, but an increase in cell death " to

When ciliogenesis is inhibited autophagy is not observed,...

6)Please correct the error: line 1198-1199

The main function that has defined the Nek family is the control of the cell cycle, including mitosis and centrosome disjunction. .

Comments on the Quality of English Language

The manuscript needs grammar and orthographic corrections. 

Author Response

We thank the reviewer for the careful correction of our manuscript. We have addressed the minor comments as follows:

Minor comments:

1)Please rephrase the sentence for more clarity line 898-900 "However, dysfunctional mitochondria are also a signal to cellular response and well 898 described responses comprehend mitophagy and unfolded protein response pathways 899 activation [163,164] Both processes constitute the mitochondrial quality control (MQC). "

We agree that the sentence was confuse and we have rephrased that for more clarity.

2) Please correct the sentence line 902: Autophagy is a complex process that involved several steps and multiple protein complexes. 

Autophagy is a complex process that involves several steps...

We have corrected the word.

3)Please rephrase for more clarity the paragraph line 912-line 924: "Autophagy is a vastly studied response of the cell to mitochondrial damage and is named mitophagy......"

Autophagy is not a response specific only for mitochondria, but also for other organelles. Authors define mitophagy twice in this paragraph

We have rephrased the sentence.

4) Keep the same symbols for UPRET and UPRmt throughout the text

We have revised that.

5)Please rephrase the sentence line 1145." When ciliogenesis is inhibited no autophagy is observed, but an increase in cell death " to

When ciliogenesis is inhibited autophagy is not observed,...

Thank you for the correction, we have changed the sentence.

6)Please correct the error: line 1198-1199

The main function that has defined the Nek family is the control of the cell cycle, including mitosis and centrosome disjunction.